# Understanding Chain-of-Thought in LLMs through Information Theory

**Jean-François Ton** [* 1]  **Muhammad Faaiz Taufiq** [* 1]  **Yang Liu** [2]

## Abstract

Large Language Models (LLMs) have shown impressive performance in complex reasoning tasks through the use of Chain-of-Thought (CoT) reasoning, allowing models to break down problems into manageable sub-tasks. However, existing CoT evaluation techniques either require annotated CoT data or fall short in accurately assessing intermediate reasoning steps, leading to high rates of false positives. In this paper, we formalize CoT reasoning in LLMs through an information-theoretic lens. Specifically, our framework quantifies the 'information-gain' at each reasoning step, enabling the identification of failure modes in LLMs without the need for expensive annotated datasets. We demonstrate the efficacy of our approach through extensive experiments on toy arithmetic, GSM8K and PRM800k datasets, where it significantly outperforms existing outcome-based methods by providing more accurate insights into model performance on individual subtasks.

## 1. Introduction

Large Language Models (LLMs) have demonstrated remarkable capabilities across a wide range of tasks, from complex reasoning to code generation (Chowdhery et al., 2024; OpenAI, 2024; Bubeck et al., 2023; Anil et al., 2023). Many of these advances can be attributed to Chain-of-Thought (CoT) reasoning (Wei et al., 2024; Nye et al., 2021; Li et al., 2024), which involves breaking down complex problems into a series of intermediate steps, mirroring human-like reasoning processes. The success of CoT reasoning, particularly in domains such as mathematics, logic, and multi-step decision-making, has led researchers to incorporate CoT-like features directly into model training, i.e. the FLAN family of models (Chung et al., 2022; Wei et al., 2022).

This paper introduces a new formal framework for analyzing CoT in LLMs. We provide a rigorous method grounded in information theory, to evaluate the quality of each step in a model's reasoning process, thus offering insights beyond simple accuracy metrics to identify areas for improvement.

Previous work in this area has proposed "*Process Supervision*" (Lightman et al., 2023), which requires expensive, human-annotated step-by-step data. While effective, this approach is often impractical due to the high cost and effort of creating large-scale annotated datasets. In turn, alternative methods have recently been proposed, such as outcome reward modelling (Havrilla et al., 2024) or the Math-Shepherd (Wang et al., 2024b). Both these approaches avoid reliance on costly annotated step-wise CoT data by instead modelling the correctness of each step based on the correctness of final outputs. However, as we show later, these methods can be unsound for detecting incorrect reasoning steps and can thus lead to a high false-positive rate in certain scenarios.

To address these shortcomings, we employ an information-theoretic approach, grounded in the following key insight: *Each correct step in a reasoning process should provide valuable and relevant information that aids in predicting the final correct outcome*. Building on this insight, we develop a framework to quantify the "*information-gain*" after each sub-task in the reasoning process, without the need for step-by-step annotations. This enables us to detect sub-tasks that fail to contribute meaningful information toward the correct solution, signalling potential errors or irrelevant steps in the model's reasoning. In addition, we also introduce a practical algorithm to assess LLM performance across various sub-tasks within a Chain-of-Thought (CoT) reasoning process. The key contributions of this paper are as follows:

1. We develop a framework for sequential applications of sub-tasks, e.g. Chain-of-Thought and provide a rigorous language to identify failure modes in LLMs.

2. Based on this framework, we propose a practical algorithm to assess the task-wise performance of models. This yields more granular information about a model's CoT performance without requiring annotated data for intermediate reasoning steps.

3. We validate our methods on extensive toy data, the GSM8K (Cobbe et al., 2021) as well as the PRM800K

* denotes equal contribution, where ordering was determined through a coin flip. [1]ByteDance Seed [2]UC Santa Cruz. Correspondence to: Jean-François Ton <jeanfrancois@bytedance.com>, Muhammad Faaiz Taufiq <faaiz.taufiq@bytedance.com>.

*Proceedings of the 42ⁿᵈ International Conference on Machine Learning*, Vancouver, Canada. PMLR 267, 2025. Copyright 2025 by the author(s).

(Lightman et al., 2023) dataset. Our method effectively identifies failure modes in CoT reasoning, unlike baselines like outcome reward modelling (Havrilla et al., 2024) and Math-Shepherd (Wang et al., 2024b), which rely on final accuracy and tend to increase false positives in error detection.

## 2. Proposed Framework: Setup and Notation

Before diving into our framework, we first provide a high-level overview and notation on how LLM generation will be treated throughout this paper. This will allow us to set the foundation for describing our information-theoretic framework. In particular, following the approach in (González & Nori, 2023), we view LLMs as abstract execution machines with a natural language interface. From this perspective, prompts are designed to solve specific problems (e.g., mathematical or logical problems), and the LLM processes the information in the prompt to generate an output.

We define a typical prompt as a combination of two parts:

1. An initial state, represented by the random variable $X_0 \in \mathcal{X}$, encapsulates the prompt-provided information that the LLM processes to derive the queried result.

2. A task $\lambda \in \Upsilon$ (e.g., addition then multiplication) defines how the LLM processes $X_0$.

Given the prompt, defined as a tuple $(X_0, \lambda)$, the state $X_1$ represents the result of applying task $\lambda$ to the initial state $X_0$. Formally, we denote this using the *update* mapping $\Lambda : \mathcal{X} \times \Upsilon \to \mathcal{X}$ which outputs the updated state $X_1$ by applying the task $\lambda$ on $X_0$, i.e. $X_1 = \Lambda(X_0, \lambda)$. This updated state is then used to obtain the final output, denoted by $Y \in \mathcal{X}$, by extracting only the information in $X_1$ which is relevant to the queried final answer. This notation defines a prompt that instructs a model to process information drawn from some initial distribution $p(X_0)$ (e.g., math problems).

We use a simple example to illustrate this notation:

**Prompt:** *"Solve for $z = 2\,(u + v)$, $u = 12$, $v = 13$"* (1)

Here, the initial state $x_0$ denotes the information *"u = 12, v = 13"*, and $\lambda$ denotes the task of finding $z$ (i.e. addition followed by multiplication). Next, $x_1 = \Lambda(x_0, \lambda)$ represents the updated information after correctly performing the addition operation, i.e. $x_1$ is *"u = 12, v = 13 and z = 50"*. The final output, $y$, is then obtained by simply extracting the value of $z$ from $x_1$, i.e. *"z = 50"*.

**Remark** Our setup also encapsulates cases with ambiguous (or multiple correct) responses for a given task $\lambda$. In this case, $\Lambda(x_0, \lambda)$ is a random variable with distribution $p(X_1 \mid X_0 = x_0)$. Therefore, for generality, we treat $\Lambda(x_0, \lambda)$ as a random variable from now on.

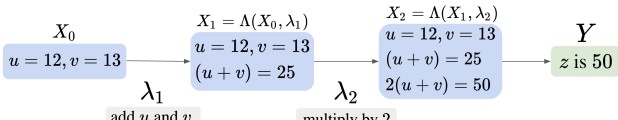

*Figure 1.* Prompt (1) requires compositional application of tasks.

### 2.1. Compositionality

Many mathematical or logical problems, such as the one in (1), require sequential application of operations. Our notation is also amenable to such problems as it accommodates the composition of tasks.

For example, one way to address prompt (1) involves first adding $u$ and $v$, and next, multiplying the result by 2 to find $z$. Using our notation, this can be expressed as $\Lambda(x_0, \lambda_1 \circ \lambda_2)$, where $\lambda_1, \lambda_2$ denote the addition and multiplication tasks respectively. The following property allows us to define the application of compositional task $\lambda_1 \circ \lambda_2$:

**Definition 2.1.** We say that an update rule $\Lambda : \mathcal{X} \times \Upsilon \to \mathcal{X}$ is *compositionally consistent* if, for all $x_0 \in \mathcal{X}$ and $\lambda_1, \lambda_2 \in \Upsilon$ we have that $\Lambda(x_0, \lambda_1 \circ \lambda_2) \overset{\mathrm{d}}{=} \Lambda(\Lambda(x_0, \lambda_1), \lambda_2)$.

Here, $\overset{\mathrm{d}}{=}$ denotes equality in distribution and is sufficient in cases where a query may have multiple correct responses.

Returning to the prompt in (1), Figure 1 shows that the model first computes $u + v$, then multiplies the result by 2. Here, we refer to $X_1, X_2$ as *intermediate* states and $Y$ is the correct final output. In general, if a problem statement requires sequential application of $T$ sub-tasks, $\lambda = \lambda_1 \circ \ldots \circ \lambda_T$, then the Chain-of-Thought (CoT) reasoning is divided up into $T$ steps, where the $t$'th step is recursively defined as $X_t = \Lambda(X_{t-1}, \lambda_t)$ for $t \in \{1, \ldots, T\}$. Finally, the overall true output $Y$ is obtained by extracting the queried information from the final state $X_T$.

Having established a formal language for the sequential application of tasks, we now turn towards how a task may be divided into such a sequence of intermediate sub-tasks.

### 2.2. Primitive Tasks

In this subsection, we introduce the notion of *primitive tasks* which form the basic building blocks of any task. Intuitively, our formulation is reminiscent of ideas from linear algebra, where basis vectors form the basic building blocks of a vector space. In our case, any task $\lambda \in \Upsilon$ can be expressed as a sequence of primitive tasks. This decomposition will allow us to establish which tasks the model could have learned from the training data. For example, if a specific primitive task is not available in the LLM training data, it would be impossible for the model to execute any instructions which involve this primitive task correctly. With this in mind, we

now introduce this concept formally:

**Definition 2.2** (Primitive tasks). We say that a set of tasks $\Gamma \subseteq \Upsilon$ is primitive if, for any task $\lambda \in \Upsilon$, there exists a unique subset $\{\lambda_i\}_{i=1}^k \subseteq \Gamma$ such that $\lambda = \lambda_1 \circ \cdots \circ \lambda_k$.

Note that the decomposition is not unique but the set of components is. In some cases, there may exist distinct permutations of primitive tasks which compose to yield the same task as is common in many associative operations. As an example, in the context of mathematical problem-solving, the basic arithmetic operations could be considered primitive. The composition of these primitive tasks allows us to construct extremely complex operations. Just like in linear algebra, we define the span of these tasks as the set obtained by their sequential applications.

**Definition 2.3** (Span of tasks). Let $\Phi \subseteq \Upsilon$ be a set of tasks:

$$\text{Span}(\Phi) = \{\lambda_1 \circ \ldots \circ \lambda_k : \lambda_i \in \Phi \text{ for } 1 \le i \le k, k \in \mathbb{Z}_{>0}\}.$$

The set $\text{Span}(\Phi)$ comprises all the tasks that can be applied by composing sub-tasks in the set $\Phi$. This means that any *compositionally consistent* update rule $\Lambda$ which is well-defined on the set of tasks $\Phi$ will also be well-defined on $\text{Span}(\Phi)$. However, this $\Lambda$ may still be ill-defined for any task not in this span. This limitation, known as unidentifiability, defines the boundaries of a model's inferences.

### 2.3. Unidentifiability

The unidentifiability of tasks forms a key part of our framework. It directly addresses the fundamental challenge that models, such as LLMs, face when dealing with unseen tasks. If a task $\lambda$ lies outside of $\text{Span}(\Phi)$, the span of tasks the model has been trained on, then the model cannot be expected to infer or apply it correctly. In other words, the model's capacity is constrained by the identifiability of tasks within the training set. This notion and formalization of unidentifiability allows us to highlight a critical limitation in the generalization of models: tasks not encountered during training cannot be reliably executed, as they remain beyond the model's learned task-span. More formally:

**Definition 2.4** (Unidentifiability). A task $\lambda$ is unidentifiable in a set $\Phi \subseteq \Upsilon$ if and only if $\lambda \notin \text{Span}(\Phi)$.

**Remark** In practice, unidentifiability may depend on the initial state $X_0$, i.e. an LLM might accurately perform addition for 2-digit numbers but fail with 10-digit numbers (Razeghi et al., 2022). For more details, see Appendix B.1.

Building on this framework, we propose an algorithm that integrates unidentifiability with information-theoretic methods to detect CoT reasoning failures.

## 3. Operationalising Our Framework

This section aims to operationalise the above framework to make inferences regarding the unidentifiability of intermediate sub-tasks in a model's CoT reasoning process. This would subsequently allow us to detect any sub-task at which a model's CoT reasoning process starts to diverge from the ground truth, thereby providing insights into how the model can be improved. For example, suppose we are in a setting where the "addition" operation is unidentifiable, then we could further improve the model's mathematical reasoning by fine-tuning it on the addition operation.

### 3.1. An information-theoretic perspective

To apply unidentifiability in CoT generations, we introduce a fundamental assumption: each correct CoT step should provide meaningful information aiding the prediction of $Y$. If a step ceases to increase information about $Y$, it indicates an incorrect execution. We formalize this assumption using our notation from the previous section:

**Assumption 3.1** (Bayesian network). Let $\lambda \ne \lambda'$ be two operations with primitive decompositions:

$$\lambda = \lambda_1 \circ \ldots \lambda_{k-1} \circ \lambda_k \circ \cdots \circ \lambda_T \quad \text{and}$$
$$\lambda' = \lambda_1 \circ \ldots \lambda_{k-1} \circ \lambda_k' \circ \cdots \circ \lambda_{T'}',$$

where $\lambda_k'$ is unidentifiable in $\{\lambda_1, \ldots, \lambda_T\}$. Then, the intermediate states corresponding to the tasks $\lambda, \lambda'$ have the Bayesian network in Figure 2.

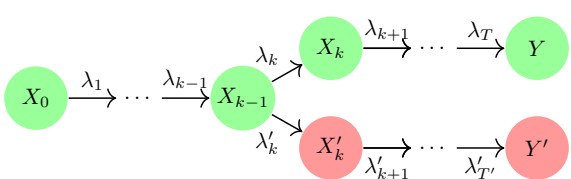

*Figure 2.* Bayesian network corresponding to Assumption 3.1.

**Intuition** The Bayesian network in Figure 2 implies that if we encounter an unidentifiable task ($\lambda_k'$) at step $k$ of the reasoning path, the future states $X_i$ and $X_j'$ for any $i, j \ge k$ satisfy the conditional independence $X_i \perp\!\!\!\perp X_j' \mid X_{k-1}$. Consequently, once we apply $\lambda_k'$, the subsequent states along the new reasoning path (in red) add no information regarding the subsequent states or the output of the original path (in green). Hence the figure represents the fact that, for any given input, the output of $\lambda_k$ (top fork) contains no information regarding the output of any other primitive task $\lambda_k'$ (bottom fork).

With our key assumption on the ground-truth CoT process formalized, we now consider the model's CoT behaviour.

## 3.2. Task execution in LLMs

To operationalise our framework, we formally distinguish between the model i.e. LLM's task execution and the *ground truth* process which arises from following the instructions correctly. To this end, we explicitly define how an LLM interprets a specified task $\lambda$ using $\Lambda^M(X_0, \lambda)$, which is in general distinct from the *ground truth* update rule $\Lambda(X_0, \lambda)$.

Here, one option would be to consider the idealised setting where the model learns to perfectly follow some of the primitive tasks available in the training data. However, this may be considered too restrictive since in reality most LLMs do not always follow a "learned" task perfectly. Instead, we consider a much weaker assumption that the model cannot correctly execute a task which is unidentifiable in the training data. Concretely, suppose $\Gamma^M \subseteq \Gamma$ denotes the primitive tasks available in the LLM training data. Then, we make the following assumption on LLM's task execution.

**Assumption 3.2** (Task execution in LLMs). $\Lambda^M$ is compositionally consistent and for any $(x_0, \lambda) \in \mathcal{X} \times \Upsilon$, there exists some $\widehat{\lambda} \in \text{Span}(\Gamma^M)$ such that $\Lambda^M(x_0, \lambda) \stackrel{\text{d}}{=} \Lambda(x_0, \widehat{\lambda})$.

**Intuition**  Assumption 3.2 means that for any task which we would like the LLM to apply, the LLM ends up executing some task in $\text{Span}(\Gamma^M)$ which the model has been trained on. In other words, the model's execution is restricted only to the tasks which could be inferred from the training data (i.e. in $\text{Span}(\Gamma^M)$). Moreover, this assumption also allows us to encapsulate cases where the model does not follow correct instructions or does not decompose a task correctly.

Before proceeding further with our main result which will allow us to test for the unidentifiability of sub-tasks, we define some notation which we will use from now onwards. Let $\lambda = \lambda_1 \circ \ldots \circ \lambda_T$ denote a primitive decomposition of a task $\lambda$. Then, starting from an initial state $X_0$, we denote the model's intermediate states recursively as:

$$X_t^M := \Lambda^M(X_{t-1}^M, \lambda_t) \quad \text{and} \quad X_0^M = X_0.$$

Moreover, we use $Y^M$ to denote the model's final output. Next, using this notation, we present the conditional independence which must hold if the model encounters an unidentifiable intermediate task along its reasoning path.

**Theorem 3.3.** *Let $\Gamma^M \subseteq \Gamma$ denote the primitive tasks available in the training data. Let $\lambda$ be a task with decomposition $\lambda = \lambda_1 \circ \ldots \circ \lambda_T$. If $\lambda_k$ is the first task in the decomposition of $\lambda$ which is unidentifiable in $\Gamma^M$ (i.e. $k = \arg\min_t\{\lambda_t \notin \text{Span}(\Gamma^M)\}$). Then, under Assumptions 3.1 and 3.2, we have that*

$$Y \perp\!\!\!\perp X_j^M \mid X_{j-1}^M \quad \text{for all } j \geq k. \tag{2}$$

Theorem 3.3 shows that under Assumptions 3.1 and 3.2, when the model encounters an unidentifiable task (i.e. $\lambda_k$ in

Theorem 3.3) in its Chain-of-Thought reasoning, the model output satisfies the conditional independence in Equation (2). In practice, this means that if at step $k$, a model encounters a reasoning step which is *necessary* for obtaining the correct answer and is *unidentifiable* in the training data, the CoT reasoning diverges from the ground truth at this step and every subsequent step adds no additional information regarding the correct final output Y. This 'information' can be measured by checking if the model's confidence about the final output $Y$ increases after each step. This is formalised in the next section.

## 3.3. Testing for unidentifiability using information-gain

With our framework established, we now describe how to detect unidentifiable sub-tasks using information theory. Following common practice (Wang et al., 2024b; Havrilla et al., 2024), we assume access to a dataset of prompts and correct final answers, derived by applying task $\lambda$, denoted as $\mathcal{D}_\lambda := \{(x_0^i, y^i)\}_{i=1}^n$. Recall that $X_j^M$ and $X_{j-1}^M$ represent the model's chain of thought (CoT) reasoning at steps $j$ and $j-1$, respectively. Consequently, each element in the conditional independence statement in Equation (2) can be derived from the data and/or the model.

To this end, we consider the mutual information between $Y$ and $X_j^M$ conditional on $X_{j-1}^M$, denoted by $\mathcal{I}(Y; X_j^M \mid X_{j-1}^M)$. This conditional mutual information term intuitively represents the *additional information* contributed by the $j$'th step of CoT, which is relevant for predicting the ground truth final output $Y$. Therefore, we refer to $\mathcal{I}(Y; X_j^M \mid X_{j-1}^M)$ as the *information-gain* at step $j$.

It follows from Theorem 3.3 that if an LLM encounters a sub-task at step $i$ which is unidentifiable in its training data, no subsequent step should contribute any additional information relevant for predicting $Y$ (i.e. the information-gain should remain 0 after step $i$). If, on the other hand, we observe that $\mathcal{I}\left(Y; X_j^M \mid X_{j-1}^M\right) > 0$ for some $j \geq i$, then under Assumptions 3.1 and 3.2, the task $\lambda_i$ is not unidentifiable. To estimate the information-gain in practice, we use the following result:

**Proposition 3.4.** *Let $\mathcal{I}(X; Y \mid Z)$ denote the mutual information between $X$ and $Y$ conditional on $Z$. Then,*

$$\mathbb{E}[\log p(Y \mid X_j^M)] - \mathbb{E}[\log p(Y \mid X_{j-1}^M)]$$
$$= \mathcal{I}\left(Y; X_j^M \mid X_{j-1}^M\right) \geq 0. \tag{3}$$

To estimate the information-gain in (3) using Proposition 3.4, we train a separate LLM, which we refer to as the *supervisor model* $g_{\text{sup}}$. This model takes as input the model's CoT reasoning up to any given intermediate step $t$, $X_t^M$, and is fine-tuned to directly predict the ground truth final output $Y$. In this way $g_{\text{sup}}(X_t^M)$ approximates the conditional distribution $p(Y \mid X_t^M)$. Then, the quantity $\mathbb{E}[\log p(Y \mid$

$X_j^M)]$ can be estimated using the negative cross-entropy loss for predicting $Y$, i.e., $\mathbb{E}[\log p(Y \mid X_j^M)]$ is approximately

$$\mathbb{E}[\log \hat{p}(Y \mid X_j^M)] = -\mathbb{E}[l_{\text{CE}}\left(Y, g_{\text{sup}}(X_j^M)\right)],$$

where $l_{\text{CE}}$ denotes the cross-entropy loss. From this, it follows that

$$\underbrace{\mathbb{E}[\log p(Y \mid X_j^M)] - \mathbb{E}[\log p(Y \mid X_{j-1}^M)]}_{\text{Information-gain}}$$
$$\approx \mathbb{E}[l_{\text{CE}}(Y, g_{\text{sup}}(X_{j-1}^M))] - \mathbb{E}[l_{\text{CE}}(Y, g_{\text{sup}}(X_j^M))]. \quad (4)$$

**Summary** The *information-gain* (IG) between steps $j$ and $j-1$ reflects how much relevant information step $j$ contributes towards predicting $Y$. If task $\lambda_j$ is executed correctly, this gain is positive, as indicated by a decrease in the cross-entropy loss. Conversely, if step $j$ does not provide additional information, the loss remains unchanged. This can be interpreted as the conditional mutual information between $X_j^M$ and $Y$, conditioned on $X_{j-1}^M$. Positive information-gain suggests step $j$ adds new insight about $Y$, while no gain indicates no added information. Training details for the supervisor model are in Appendix C.1.3.

**Remark on sample-wise information-gain** While conditional mutual information provides an aggregate measure of information-gain for a sub-task in a dataset, it may also be desirable to obtain an analogous measure of sub-task correctness for individual CoT instances. This could be useful, for example, in detecting which step is wrong for a given prompt. Our notion of information-gain can be extended to this sample-wise setting, similar to (Ethayarajh et al., 2022), by instead considering the following difference

$$\log p(Y \mid X_j^M) - \log p(Y \mid X_{j-1}^M)$$
$$\approx l_{\text{CE}}(Y, g_{\text{sup}}(X_{j-1}^M)) - l_{\text{CE}}(Y, g_{\text{sup}}(X_j^M)). \quad (5)$$

Intuitively, if step $j$ in the model's CoT is correct, the model should become more confident in the ground truth output $Y$ being the correct final answer. Therefore, the difference above should be positive. Conversely, if step $j$ is wrong, the model's confidence regarding $Y$ should not increase, and this difference should be $\leq 0$. From now on, we refer to the difference in (5) as *sample-wise information-gain* at step $j$.

**Remark on O1/R1 style reasoning** Although we present our framework using linear chains-of-thought for clarity, the information-gain metric naturally accommodates the more complex reasoning patterns found in O1/R1-style models. These reasoning models often explore multiple solution paths, backtrack when encountering errors, and dynamically self-correct their approach. In such exploratory settings, our framework remains effective: steps along incorrect reasoning trajectories will exhibit low or negative information gain, indicating they do not contribute meaningfully toward

the correct final answer. When the model identifies a more promising path and self-corrects, subsequent steps will show positive information gain, signaling productive progress.

This adaptability to varied reasoning structures is empirically demonstrated in our experiments (Section 5), where we analyse reasoning traces from the MATH/PRM dataset. Steps that human annotators labelled as uninformative or irrelevant consistently show low information gain under our metric, while correct and meaningful steps exhibit high information gain. Thus, our method provides reliable step-wise evaluation regardless of whether the reasoning follows a linear chain or involves branched/exploratory patterns.

We further formalize this remark using our framework in Appendix B.4.

# 4. Related Works

**Evaluation of CoT reasoning** Several recent works propose methodologies for evaluating CoT reasoning (Wei et al., 2024; Havrilla et al., 2024; Li et al., 2023; Joshi et al., 2023; Nguyen et al., 2024; Wang et al., 2024a; Yu et al., 2024; Xie et al., 2024). For example, (Li et al., 2023) verifies individual steps in a model's CoT reasoning by generating multiple LLM responses per prompt and comparing correct responses with incorrect ones.

Similarly, (Wang et al., 2024b;c) use a fine-tuned LLM to decode multiple reasoning paths from each step and check the correctness of these reasoning paths. However, as we show in our experiments, approaches which simply rely on the correctness of the final output are not sound in general and can lead to false positives. Moreover, these solutions may not be plausible for problems of high difficulty where correct LLM responses might be scarce.

**Formalising CoT framework** The formalisation of LLM reasoning remains an active area of research. Most notably (González & Nori, 2023) introduces a formal framework for LLMs and is a key source of inspiration behind our formalism. Additionally, (Feng et al., 2023) theoretically examines the expressivity of LLMs with CoT in solving mathematical and decision-making problems, focusing on the transformer architecture's implications on accuracy. Besides this, (Xu et al., 2024) provides a formal definition of hallucinations, but does not consider CoT reasoning specifically.

**Reward modelling** Outcome-based reward models (ORM) (Cobbe et al., 2021; Havrilla et al., 2024; Lightman et al., 2023) predict the probability of reaching the correct final answer based on a model's intermediate CoT steps. While they avoid requiring correct intermediate demonstrations, we show in Section 5 that they are unsound for detecting CoT reasoning errors. Step-wise ORM (SORM) (Havrilla et al., 2024) extends ORM by estimating the probability of

an 'optimal' model reaching a correct answer but requires training a larger, more capable model than the base model.

Process-based reward modelling (PRMs) (Lightman et al., 2023; Uesato et al., 2022) is an alternative approach which directly predicts the correctness of intermediate CoT reasoning steps. Likewise, various other approaches rely on annotated CoT datasets for benchmarking (Jacovi et al., 2024; Yu et al., 2024; Amini et al., 2019; Liu et al., 2020; Xi et al., 2024; Nguyen et al., 2024; Xie et al., 2024; McLeish et al., 2024). While these benchmarks and methodologies aid LLM reasoning, collecting annotated data is costly and not easily scalable. In contrast, our approach evaluates an LLM's CoT reasoning without human-annotated CoT data.

# 5. Experiments

In this section, we demonstrate our framework's utility, dubbed Information-Gain (IG) and compare against two baselines for detecting errors in a model's CoT reasoning. Here we assume access only to the model's CoT generations $X_0, X_1^M, \ldots, X_T^M$ and correct final answers $Y$.

**Outcome Reward Model (ORM) (Havrilla et al., 2024)** This involves training a classifier, denoted as $f_{\text{ORM}}$, which takes as input model generations up to any step $t$ in its CoT reasoning, $X_t^M$, and predicts the probability of the model's final answer being correct, i.e.

$$f_{\text{ORM}}(X_t^M) \approx \mathbb{P}(Y^M = Y \mid X_t^M). \quad (6)$$

Here, if we observe that this probability of correctness drops significantly after step $t$, i.e. $f_{\text{ORM}}(X_t^M) \gg f_{\text{ORM}}(X_{t+1}^M)$, this indicates that the model applies task $\lambda_{t+1}$ incorrectly.

**Math-Shepherd (MS) (Wang et al., 2024b)** This method quantifies the *potential* for a given reasoning process $X_t^M$ by using a 'completer' model to generate $N$ completions of each reasoning process starting from step $t$, $\{(X_t^M, X_{t+1,j}^M, \ldots, X_{T,j}^M, Y_j^M)\}_{j \leq N}$, where $Y_j^M$ denotes the final answer reached in the $j$'th completion. Then, we estimate the potential of this step based on the proportion of correct answers among the $N$ completions:

$$f_{\text{MS}}(X_t^M) := \sum_{j=1}^N \mathbb{1}(Y_j^M = Y)/N. \quad (7)$$

For a fair comparison, we do not assume access to a 'verifier' model more capable than our base model. Therefore, we use the base model as the completer model in our experiments.

## 5.1. Toy data experiments

We first consider a toy setting where we control model behaviour across tasks. Prompts consist of an integer vector $Z_0 \in \mathbb{Z}^5$ sampled from a given distribution. The task $\lambda$

comprises five steps, $\lambda = \lambda_1 \circ \ldots \circ \lambda_5$, where each sub-task $\lambda_i$ transforms $Z_{i-1} \in \mathbb{Z}^5$ into $Z_i \in \mathbb{Z}^5$. The correct final answer $Y$ is $Z_5$. Additional details on data generation and sub-tasks are in Appendix C.1.

**Generating the dataset** To investigate partial unidentifiability for a given task $\lambda_i$ we modify the obtained dataset by introducing 'noise' at step $i$. In other words, the task $\lambda_i$ is applied incorrectly on a subset of the data, whereas all other tasks are always applied correctly. This represents a model which sometimes fails at step $i$ and we use 'LLM$_i$' to denote this model in this experiment. We repeat this procedure for all tasks $\lambda_i$ for $i \in \{1, \ldots, 5\}$ which yields 5 LLMs $\{\text{LLM}_1, \ldots, \text{LLM}_5\}$.

To assess robustness, we introduce a special case in LLM$_3$, where task $\lambda_3$ is applied incorrectly iff the output after $\lambda_2$ falls in a set $\mathcal{S}$. This deliberate choice highlights a pitfall of existing baselines and contrasts with other LLMs, where errors occur randomly. In other words, $\lambda_3$'s correctness depends on $\lambda_2$'s output. For details, see Appendix C.1.2.

### 5.1.1. RESULTS

Figure 3 shows how the different baselines quantify the correctness of the different tasks for the 5 different LLMs under consideration. This figure only considers samples where the final answer of the LLM was incorrect, i.e. $Y^M \neq Y$. For our method (IG), Figure 3a shows the information-gain across the different steps for each LLM. Likewise, Figure 3b presents the results for ORM and shows how the average probability of correctness in (6) changes across the different steps, whereas, for Math-Shepherd, Figure 3c shows the proportion of correct completions starting after each step (7). Here, any significant drop in the plotted values indicate an incorrect application of a task.

**Information-gain rightly quantifies step-wise correctness** We observe that the information-gain remains positive for each LLM until we encounter an incorrect reasoning step, at which point it drops to negative values. Therefore, our method can identify the incorrectly executed task for each LLM under consideration. We used a GPT-2 supervisor model to estimate information-gain.

**Pitfall of the baselines** While ORM and Math-Shepherd usually identify incorrect reasoning steps, they fail for LLM$_3$. This is because $\lambda_3$ is misapplied iff the output after $\lambda_2$ lies in $\mathcal{S}$. Thus, the classifier can predict the final output's correctness at $\lambda_2$ by checking if $Z_2$ lies in $\mathcal{S}$, leading to overconfidence in error detection at $\lambda_2$ instead of $\lambda_3$.

Similarly, with Math-Shepherd for LLM$_3$ (using the same model as a completer), a completion is incorrect if the output after $\lambda_2$ lies in $\mathcal{S}$. Here, all completions fail, regardless of the starting step, making it impossible to pinpoint where LLM$_3$ goes wrong.

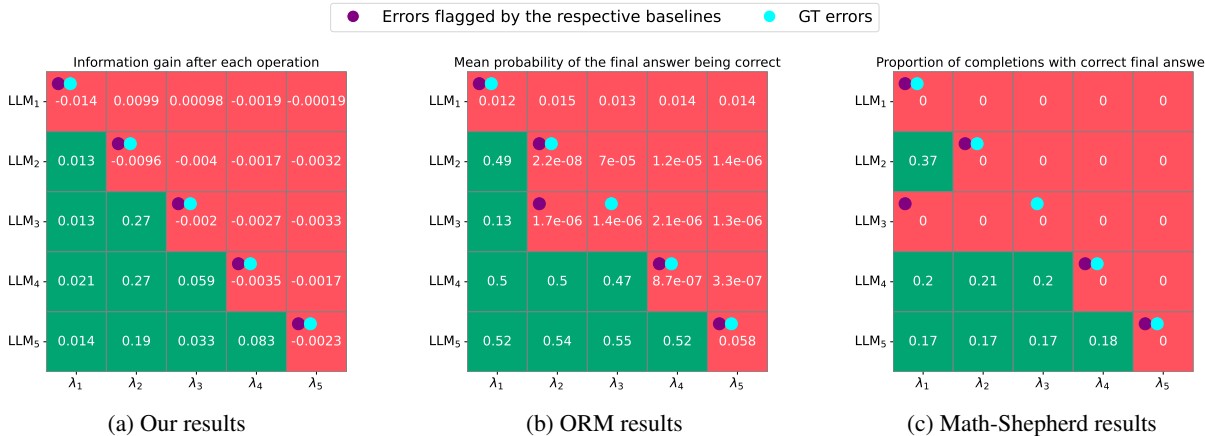

(a) Our results      (b) ORM results      (c) Math-Shepherd results

*Figure 3.* Heatmaps (left) quantifying the correctness of different sub-tasks for the 5 LLMs using the different baselines, and the associated classification metrics (right). Red color in the heatmaps indicates a significant drop in the plotted metrics (an incorrectly executed sub-task).

*Table 1.* Sample-wise classification of a sub-task for $LLM_3$.

|             | ACC ↑ | TPR ↑ | FPR ↓ |
|-------------|-------|-------|-------|
| IG (OURS)   | **0.96** | 0.98 | **0.06** |
| ORM         | 0.77  | 0.98  | 0.54  |
| MS          | 0.60  | **1.0** | 1.0 |

**Sample-wise detection** We also use the different baselines for sample-wise detection of erroneous steps, as outlined in Section 3.3. A step is classified as incorrect if a baseline's metric falls below a threshold. Table 1 presents the results for $LLM_3$, with optimal thresholds chosen from a held-out dataset. Our method achieves significantly higher accuracy and fewer false positives than the baselines, making it more reliable for sample-wise error detection.

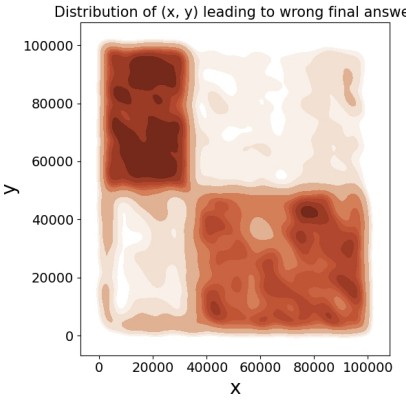

*Figure 4.* The distribution of $(x, y)$ for incorrect samples: Llama-3-8B struggles to add large and small numbers (represented by the top-left and bottom-right shaded regions).

## 5.2. Arithmetic operations on Llama-3-8B

Following our toy experiments, we now evaluate our framework in a more realistic setting using the Llama-3-8B model

(Dubey et al., 2024). We focus on a simple arithmetic task that involves both multiplication and addition. The goal is to assess the model's performance on each operation.

**Experimental setup** We sample two integers $x$ and $y$ uniformly from $[1, 10^5]$. The prompt to the model is:

**Prompt:** "$x = \{x\}$, $y = \{y\}$, *Please calculate the following: 1. 3x, 2. 2y, 3. 3x + 2y* "

**Model Accuracy** The model's accuracy across steps is:

Step 1: 80%,     Step 2: 98%,     Step 3: 42%.

Most failures occur in Step 3, which involves adding previously computed values. Analyzing the $(x, y)$ distribution where the model is incorrect (Figure 4), we find that errors mainly arise when one variable is large and the other is small. This suggests that correctness depends heavily on $(x, y)$, making it difficult for baselines to pinpoint the erroneous step in the model's CoT reasoning.

### 5.2.1. RESULTS

**Our method** We trained the supervisor model by fine-tuning a Llama-3-8b model using Low Rank Adaptation (LoRA) (Hu et al., 2021). Table 2 shows that there is a significant drop in information-gain at step 3 relative to steps 1 and 2, demonstrating that our method correctly identifies that the failure mainly occurs at step 3.

**Outcome Reward Model (ORM)** For ORM, the mean probability of correctness (Table 2) remains unchanged at each step. Figure 4 suggests this is because ORM predicts correctness based solely on $x$ and $y$ in the prompt. Crucially, its confidence remains constant even as intermediate reasoning steps are added, preventing it from distinguishing the model's performance at different steps.

**Math-Shepherd (MS)**    Table 2 shows the proportion of correct completions for MS. While this is low at step 3, only 5-7% of completions from steps 1 and 2 yield a correct output, despite errors mostly occurring at step 3. This is because Llama-3-8B's correctness is largely determined by $(x, y)$ in the prompt (Figure 4). As a result, MS frequently mislabels steps 1 and 2 as incorrect, leading to a higher false positive rate compared to our baseline.

**Sample-wise detection**    When using these methods for sample-wise detection of incorrect steps, our approach yields the highest accuracy among the baselines considered. This superior performance is attributed to the fact that baselines like ORM and MS often falsely flag steps 1 and 2 as incorrect, as evidenced by their high FPRs in Table 2.

### 5.3. Experiments on the controlled GSM8K Dataset

To evaluate our method on a complex dataset, we conducted experiments on GSM8K (Cobbe et al., 2021), controlling specific factors for more interpretable results.

We begin by using GPT-4 (OpenAI, 2024) to generate answers for GSM8K questions where the "multiplication" operation is always done incorrectly, while all other operations are correct. Next, we filtered the dataset to ensure that "multiplication", "subtraction", and "addition" never appeared together within the same Chain of Thought (CoT) solution. In particular, we ensured in our setting that, all incorrect final answers included both "multiplication" and "subtraction", whereas correct final answers did not involve either operation. This introduces a spurious correlation between "subtraction" and wrong answers.

In this setup, we mainly focused on evaluating ORM and our proposed method, as MS (with the same completer) fails trivially under these conditions. Specifically, "multiplication" is inherently unidentifiable, since any CoT containing "multiplication" negates the influence of other sub-tasks by design. Further details on the experimental setup can be found in Appendix C.3.

#### 5.3.1. RESULTS

Table 2 shows that our information-theoretic approach (IG) successfully identifies the unidentifiable sub-task. Since the "multiplication" rule is intentionally incorrect, it yields minimal to no information gain, as expected. However, ORM results reveal a different pattern: both "multiplication" and "subtraction" have low correctness probabilities, as they are linked to incorrect final answers. This suggests that the standard ORM approach may misleadingly classify "subtraction" as incorrect.

Additionally, in our sample-wise experiment, we observe a similar trend when we use the methods to assess the sample-wise correctness of "multiplication" and "subtraction" for

each prompt. Here, our proposed method not only accurately detects the unidentifiable sub-task but also highlights a significant shortcoming of ORM. Specifically, ORM falsely flags "subtraction", which is actually correct, as an incorrect sub-task due to spurious correlations.

### 5.4. Experiments on the PRM800K dataset

To further demonstrate the practical applicability of our method, we have conducted an additional experiment on OpenAI's PRM800k dataset (Lightman et al., 2023) which is obtained by labeling the intermediate steps of the MATH dataset (Hendrycks et al., 2021).

More specifically, this dataset is a process supervision dataset with step-level correctness labels for model-generated solutions to MATH problems. To create it, Lightman et al. (2023) asked human annotators to label each step from fine-tuned GPT-4 solutions as positive (+1), negative (-1), or neutral (0). A positive label indicates a correct, reasonable step; a negative label denotes an incorrect or unreasonable step; and a neutral label indicates ambiguity (e.g., subtly misleading or technically valid yet poor).

The objective is to identify incorrect Chain-of-Thought (CoT) steps, specifically those labelled as (-1) by annotators, using our method as well as ORM. However, we do not utilize the step-wise labels during the process; they are only used for evaluation purposes. Since the base GPT-4 model used to generate the PRM data is not publicly available, we were unable to obtain MS results for this dataset.

#### 5.4.1. RESULTS

Table 2 shows the information gain and mean correctness probability for positive, negative, and neutral sub-steps.

As expected, these results show that the information-gain is significantly lower for incorrect steps (with labels -1) than for labels +1. Additionally, we also observe that the information-gain is negative for neutral steps (with labels 0), which is explained by the fact that these steps do not add any useful information regarding the ground truth (as these were deemed irrelevant/ambiguous by the human labellers). In contrast, the average probability of correctness for the ORM classifier is roughly the same across each label and, on average, is not very informative.

**Sample-wise detection**    Additionally, we also used the sample-wise information-gain (IG) as well as the ORM baseline to classify if a step is correct (as outlined in Section 3.3). To avoid ambiguity, we filtered out the neutral sub-steps (with labels 0) for this experiment and considered a balanced held-out dataset with equal number of correct and incorrect steps. Table 2 also shows the sample-wise results for both methods (where we chose the best thresholds for each baseline using a held-out dataset).

*Table 2.* Experimental results for Toy Arithmetic, GSM8K and PRM800K experiments. In each of the datasets, we denote the "correct" and "wrong" steps with ✓ and ✗ respectively.

| DATASETS & METHODS | | OPERATIONS | | | | | SAMPLE-WISE DETECTION METRICS | | |
| --- | --- | --- | --- | --- | --- | --- | --- | --- | --- |
| | METHODS | STEP 1: $3x$ ✓ | STEP 2: $2y$ ✓ | STEP 3: $3x + 2y$ ✗ | - | | ACCURACY ↑ | TPR ↑ | FPR ↓ |
| TOY ARITHMETIC | IG (OURS) | 0.67 | 0.24 | 0.027 | - | | **0.76** | 0.51 | **0.02** |
| | ORM | 0.24 | 0.24 | 0.24 | - | | 0.56 | 0.10 | 0.07 |
| | MS | 0.068 | 0.059 | 0.00069 | - | | 0.53 | **0.99** | 0.86 |
| | METHODS | ADDITION ✓ | MULTIPLICATION ✗ | DIVISION ✓ | SUBTRACTION ✓ | | ACCURACY ↑ | TPR ↑ | FPR ↓ |
| GSM8K | IG (OURS) | 0.99 | 0.026 | 1.05 | 1.06 | | **0.72** | 0.95 | **0.62** |
| | ORM | 0.46 | 0.024 | 0.38 | 0.013 | | 0.58 | **1.00** | 1.00 |
| | METHODS | NEGATIVE ✗ | NEUTRAL ✗ | POSITIVE ✓ | - | | ACCURACY ↑ | TPR ↑ | FPR ↓ |
| PRM800K | IG (OURS) | 0.058 | -0.011 | 0.168 | - | | **0.74** | **0.84** | 0.37 |
| | ORM | 0.734 | 0.745 | 0.744 | - | | 0.69 | 0.55 | **0.18** |

It can be seen that the accuracy of our method is higher than that of the ORM classifier. Additionally, our method also leads to higher TPR (and hence a lower FNR) than the ORM classifier. These results show that our method outperforms the outcome-based baselines on more complex datasets such as the MATH data as well.

## 6. Discussion and Limitations

In this paper, we introduce a novel information-theoretic approach to evaluate Chain-of-Thought (CoT) reasoning in LLMs without annotated intermediate steps. Our framework effectively identifies erroneous reasoning across diverse settings and consistently outperforms baselines, including Outcome Reward Models (ORMs) (Havrilla et al., 2024) and Math-Shepherd (MS) (Wang et al., 2024b). However, our approach does have some limitations.

Although our method avoids human-annotated step-wise data, it requires additional training of the supervisor model, which is computationally expensive. Future work could explore in-context learning to estimate information gain, reducing training needs and improving efficiency. Additionally, while our method does not require correctness labels for every step, we still need to categorize each step according to its respective sub-task. However, this limitation is not unique to our method, as both ORM and MS also rely on such labels to draw sub-task-specific conclusions.

Lastly, while we focus on logical and mathematical datasets, our method also extends to other domains requiring CoT reasoning, such as Blocks World (Slaney & Thiébaux, 2001). As we discuss in Appendix B.2, this is an interesting avenue which we leave for future research.

## Impact Statement

This paper presents work whose goal is to advance the field of Machine Learning. There are many potential societal consequences of our work, none which we feel must be specifically highlighted here.

## Acknowledgements

We would like to thank Li Hang for his valuable insights and guidance during the development of this work.

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

# A. Proofs

*Proof of Theorem 3.3.* Suppose $\lambda$ and $\lambda'$ are two tasks with primitive decompositions

$$\lambda' = \lambda'_1 \circ \cdots \circ \lambda'_{T'}$$

and

$$\lambda = \lambda_1 \circ \cdots \circ \lambda_T, \tag{8}$$

where $\arg\min_t\{\lambda_t \notin \mathrm{Span}(\{\lambda'_1, \ldots, \lambda'_{T'}\})\} \leq k$. In other words, the primitive decompositions of $\lambda'$ and $\lambda$ diverge before step $k+1$. Then, Assumption 3.1 implies that for any $j \geq k$, we have that the answer $Y$ and $X'_j$ are d-separated by $X'_{j-1}$. Therefore,

$$Y \perp\!\!\!\perp X'_j \mid X'_{j-1}.$$

Next, we know from Assumption 3.2 that there exists some task $\hat{\lambda} \in \mathrm{Span}(\Gamma^M)$ (possibly dependent on $X_0$ and $\lambda$) such that $\Lambda^M(X_0, \lambda) \stackrel{\mathrm{d}}{=} \Lambda(X_0, \hat{\lambda})$. Suppose that $\hat{\lambda}$ has primitive decomposition

$$\hat{\lambda} = \tilde{\lambda}_1 \circ \cdots \circ \tilde{\lambda}_{\tilde{T}},$$

then since $\hat{\lambda} \in \mathrm{Span}(\Gamma^M)$, we know that $\tilde{\lambda}_i \in \Gamma^M$ for $i \in \{1, \ldots, \tilde{T}\}$. If the primitive decomposition of $\lambda$ in (8) is such that $k = \arg\min_t\{\lambda_t \notin \mathrm{Span}(\Gamma^M)\}$, then we know that $\arg\min_t\{\lambda_t \notin \mathrm{Span}(\{\tilde{\lambda}_1, \ldots, \tilde{\lambda}_{\tilde{T}}\})\} \leq k$. Then, from the above it follows that

$$Y \perp\!\!\!\perp X^M_j \mid X^M_{j-1}.$$

Here, we used the fact that $X^M_j \stackrel{\mathrm{d}}{=} \Lambda(X_0, \tilde{\lambda}_1 \circ \cdots \circ \tilde{\lambda}_j)$ using Assumption 3.2.

$\square$

*Proof of Proposition 3.4.*

$$
\begin{aligned}
\mathbb{E}[\log p(Y \mid X^M_j)] - \mathbb{E}[\log p(Y \mid X^M_{j-1})] &= \mathbb{E}\left[\log \frac{p(Y \mid X^M_j)}{p(Y \mid X^M_{j-1})}\right] \\
&= \mathbb{E}\left[\log \frac{p(Y \mid X^M_j, X^M_{j-1})}{p(Y \mid X^M_{j-1})}\right] \\
&= \mathbb{E}\left[\log \frac{p(Y, X^M_j \mid X^M_{j-1})}{p(Y \mid X^M_{j-1})\, p(X^M_j \mid X^M_{j-1})}\right] \\
&= \mathcal{I}(Y, X^M_j \mid X^M_{j-1})
\end{aligned}
\tag{9}
$$

Here, the second equality above arises from the fact that $X^M_j$ also captures all the information captured in $X^M_{j-1}$ (and possibly more). Therefore, conditional on $X^M_j$, the state $X^M_{j-1}$ is deterministic and hence, $Y \perp\!\!\!\perp X^M_{j-1} \mid X^M_j$. $\square$

## A.1. Symmetry property of information-gain $\mathcal{I}(Y, X^M_j \mid X^M_{j-1})$

The mutual information between two random variables $X$ and $Y$, $\mathcal{I}(X, Y)$, is symmetric in its arguments (i.e., w.r.t. $X$ and $Y$). However, the conditional mutual information $\mathcal{I}(Y, X^M_j \mid X^M_{j-1})$ satisfies a symmetry property *conditional* on $X^M_{j-1}$. Formally, this property of the information-gain term can be expressed as follows:

There exists some functional $\mathcal{F} : \mathcal{P} \times \mathcal{P} \times \mathcal{P} \to \mathbb{R}$ where $\mathcal{P}$ is the space of probability distributions, such that

1. The information-gain $\mathcal{I}(Y, X^M_j \mid X^M_{j-1})$ can be expressed as:

$$\mathcal{I}(Y, X^M_j \mid X^M_{j-1}) = E\left[\mathcal{F}\left(P_{Y|X^M_{j-1}}, P_{X^M_j|X^M_{j-1}}, P_{Y, X^M_j|X^M_{j-1}}\right)\right],$$

where $P_Z$ denotes the distribution of the random variable $Z$ and

2. $\mathcal{F}$ is symmetric w.r.t. its first two arguments, i.e. $\mathcal{F}(p, q, r) = \mathcal{F}(q, p, r)$. Note that there is no symmetry requirement w.r.t. the third argument of $\mathcal{F}$ because the joint distribution $P_{Y, X_j^M | X_{j-1}^M}$ is already symmetric w.r.t. $Y$ and $X_j^M$ i.e. $P_{Y, X_j^M | X_{j-1}^M} = P_{X_j^M, Y | X_{j-1}^M}$.

It follows from Eq. (9) in the proof of Proposition 3.4, that the functional $\mathcal{F}$ which satisfies the above conditions is

$$\mathcal{F}(p, q, r) = -E_{X \sim p}[\log p(X)] - E_{X \sim q}[\log q(X)] + E_{X \sim r}[\log r(X)].$$

## B. Additional details of our framework

### B.1. State-conditioned unidentifiability

In practice, the concept of unidentifiability may depend on the initial state $X_0$. For instance, an LLM might accurately perform addition for 2-digit numbers but fail with 10-digit numbers (Razeghi et al., 2022). Our framework can be extended to account for such cases by explicitly incorporating the distribution of initial states into the notion of identifiability. For example, addition could be considered unidentifiable when the initial state distribution is $p(X_0 \mid X_0$ includes 10-digit numbers). However, for simplicity, we keep this distributional dependence implicit in our framework.

### B.2. How to define steps beyond mathematical datasets

The framework presented in this paper primarily considers examples related to mathematical reasoning, where the definition of primitive tasks is intuitive and well-structured. However, our methodology could be applied to other domains where the identification of primitive tasks is less straightforward, such as Blocks World (Winograd, 1972; Slaney & Thiébaux, 2001) and commonsense question answering (QA) (Talmor et al., 2019).

In the case of Blocks World, the primary task of planning can be decomposed into sub-tasks involving sequences of primitive actions, such as "stack", "unstack" and "move". Applying the information-gain methodology in this context could provide insights into the effectiveness of large language models (LLMs) in planning and executing these sub-tasks. By analyzing the information-gain for each step, it would be possible to assess where the model's reasoning process is effective and where it encounters difficulties.

Similarly, in commonsense QA, the chain-of-thought (CoT) reasoning steps can be categorized into distinct types, such as causal reasoning (identifying cause-and-effect relationships), temporal reasoning (understanding sequences and timing), and spatial reasoning (comprehending physical arrangements and object relationships). These reasoning types align naturally with the proposed framework, enabling a systematic evaluation of the LLM's decision-making process within each category.

While these extensions present promising directions for future research, they also introduce additional challenges, particularly regarding the assumptions underlying our methodology. Addressing these challenges and validating the framework across diverse domains remains an open avenue for further investigation.

### B.3. Correct final answers with incorrect intermediate steps

A notable phenomenon of chain-of-thought (CoT) reasoning is the occurrence of cases where a model arrives at the correct final answer despite containing errors in intermediate steps. This scenario raises important questions regarding the validity of intermediate reasoning and the implications for evaluating model performance.

The methodology proposed in this paper estimates the information contributed by each successive reasoning step toward the final correct answer. In instances where an intermediate step is incorrect, it is expected that this step contributes no additional relevant information, resulting in an information-gain of zero at that point—regardless of whether the final answer is ultimately correct. Conversely, if a model systematically produces incorrect intermediate steps that nevertheless lead to the correct final answer, baseline methods such as ORM and Math Shepherd would fail to detect such errors, as these approaches primarily assess correctness based on the final output.

To better understand the prevalence of this phenomenon, we conducted an additional analysis of our datasets. In our arithmetic experiments (Section 5.2), we found that only 1.2% of samples exhibited this behavior, where an intermediate step was incorrect but the final answer remained correct. In the synthetic toy experiment, such cases did not occur, as we maintained a degree of control over the data generation process. Given the low frequency of these occurrences in our settings, their impact on the overall effectiveness of our framework is minimal.

**B.4. Clarification on Assumption 3.1 and Non-Linear Reasoning Structures**

**Overview of Assumption 3.1**    Assumption 3.1 specifically addresses a particular class of reasoning steps: those that are *necessary* for solving a problem but *unidentifiable* in the training data, meaning no composition of learned tasks can yield that step. For such unidentifiable steps, we assume that once the model diverges at this point, subsequent steps do not add further information toward the correct final output. This assumption applies only to steps that cannot be executed through any combination of the model's learned capabilities.

**Application to Reasoning Systems**    Although our framework is presented using linear chains-of-thought for clarity, the information-gain metric naturally accommodates the more complex reasoning patterns found in modern systems like O1/R1-style models. These reasoning models often explore multiple solution paths, backtrack when encountering errors, and dynamically self-correct their approach. Our framework handles such non-linear reasoning structures through its information-theoretic foundation, without requiring modifications to the core assumptions.

**Mathematical Formalization**    Consider a scenario where a model explores alternative paths during reasoning. Let the correct reasoning path to final answer $Y$ be:

$$X_0 \to X_1 \to \cdots \to X_T \to Y \tag{10}$$

If the model temporarily explores an incorrect or exploratory path through some intermediate step $Z$ at position $t$, but then returns to continue correctly, the full reasoning trace becomes:

$$X_0 \to \cdots \to X_t \to Z \to X_t \to X_{t+1} \to \cdots \to X_T \to Y \tag{11}$$

Our framework evaluates such paths through conditional mutual information. Since $Y \perp Z \mid X_t$ (the final answer is conditionally independent of the exploratory step given the state at $X_t$), the information gain at step $Z$ will be zero or negative. This correctly indicates that the exploratory step $Z$ does not contribute meaningful information toward the final answer. Once the model returns to the productive path at $X_t$, subsequent steps will exhibit positive information gain, reflecting meaningful progress toward $Y$.

**Distinction Between Unidentifiable and Exploratory Steps**    It is crucial to distinguish between two types of problematic steps:

- **Unidentifiable steps** (addressed by Assumption 3.1): Steps that are necessary for the solution but cannot be executed through any composition of the model's learned tasks. These represent fundamental gaps in the model's capabilities.

- **Exploratory or incorrect steps**: Steps where the model temporarily pursues an unproductive path but can self-correct using its existing capabilities. These steps will show low or negative information gain but do not prevent the model from eventually finding the correct solution.

Self-corrected steps demonstrate that the model possesses the necessary learned operations to eventually find the correct path, whereas unidentifiable steps represent gaps that cannot be bridged through any composition of training tasks.

**Empirical Validation**    Our experiments on the MATH/PRM dataset (Section 5) confirm this theoretical framework: steps labeled as uninformative or incorrect by human annotators consistently show low or negative information gain, while correct steps exhibit high positive information gain. This demonstrates that our method reliably evaluates both linear and non-linear reasoning patterns without requiring special handling for self-corrections or exploratory paths.

## C. Additional Experimental Details

### C.1. Toy Data Experiments

In this section, we describe the exact procedure used to generate the toy data for training and evaluating the models in our experiments. The dataset is constructed through five sequential operations (or tasks) applied to an initial state $z_0$, where each

task $\lambda_i$ generates an intermediate state $z_i$. Both **correct** and **incorrect** examples were generated, with incorrect examples created by introducing random noise or permutations into the transformations.

The data was used to represent models $LLM_1$, $LLM_2$, ..., $LLM_5$, each corresponding to a setting where a specific task $\lambda_i$ was partially corrupted to simulate an unidentifiable task for that model.

### C.1.1. DATA GENERATION TASKS

For each prompt, an initial 5-element vector $z_0$ was randomly sampled, and we use the notation $z_0[i]$ to denote the $i$'th component of this vector. Next, the following tasks were applied sequentially:

**Task $\lambda_1$: Pairwise Swapping**

- Correct Mapping: The first and second elements, as well as the third and fourth elements of $z_0$, are swapped:

$$z_1[0], z_1[1], z_1[2], z_1[3] = z_0[1], z_0[0], z_0[3], z_0[2]$$

- Incorrect Mapping: The entire vector is shuffled randomly.

**Task $\lambda_2$: Cumulative Summation**

- Correct Mapping: The first three elements of $z_1$ are replaced by their cumulative sum, and the fourth and fifth elements are swapped:

$$z_2 = [z_1[0], z_1[0] + z_1[1], z_1[0] + z_1[1] + z_1[2], z_1[4], z_1[3]]$$

- Incorrect Mapping: Each element of $z_1$ is perturbed by adding a random integer between 10 and 99:

$$z_2[i] = z_1[i] + U_i \quad \text{for each } i \text{ where } U_i \text{ is a randomly sampled integer between 10 and 99}$$

**Task $\lambda_3$: Reverse and Cumulative Sum**

- Correct Mapping: The first three elements of $z_2$ are reversed, and the last two elements are replaced by their cumulative sum:

$$z_3 = [z_2[2], z_2[1], z_2[0], z_2[3], z_2[3] + z_2[4]]$$

- Incorrect Mapping: As with task $\lambda_2$, each element of $z_2$ is perturbed by adding a random integer between 10 and 99.

**Task $\lambda_4$: Sorting and Elementwise Multiplication**

- Correct Mapping: The vector $z_3$ is sorted, and the first four elements are replaced by element-wise multiplications of specific pairs:

$$z_4[0] = z_3[1] \times z_3[2], \quad z_4[1] = z_3[0] \times z_3[3], \quad z_4[2] = z_3[4] \times z_3[0], \quad z_4[3] = z_3[2] \times z_3[2]$$

- Incorrect Mapping: The vector is randomly shuffled.

**Task $\lambda_5$: Difference Calculation**

- Correct Mapping: The first element is replaced by the absolute difference of the first two elements of $z_4$, and other elements are transformed as follows:

$$z_5 = [|z_4[0] - z_4[1]|, z_4[2], z_4[3], |z_4[3] - z_4[4]|, z_4[0]]$$

- Incorrect Mapping: The vector is randomly shuffled.

## C.1.2. MODELS $\text{LLM}_1, \text{LLM}_2, \ldots, \text{LLM}_5$

For each model $\text{LLM}_i$ ($i \in \{1, 2, 3, 4, 5\}$), the task $\lambda_i$ was selectively corrupted to simulate unidentifiability for that task. Specifically:

- Correct Data: The task $\lambda_i$ was applied according to its correct mapping.

- Incorrect Data: The task $\lambda_i$ was applied using its incorrect mapping (random noise, shuffling, or perturbations).

For each $\text{LLM}_i$, the tasks $\lambda_1$ to $\lambda_{i-1}$ and $\lambda_{i+1}$ to $\lambda_5$ were correctly applied, but task $\lambda_i$ was corrupted for a subset of the data. More specifically, for all LLMs except $\text{LLM}_3$, the error was introduced at step $i$ at random with probability 0.5. In contrast, for $\text{LLM}_3$, the error was introduced at step 3 if and only if the output at step 2, $z_2$ satisfies, $z_2[2] > 150$. This choice was deliberately made to highlight a pitfall of the baselines as explained in Section 5.

**String Representation of Chain-of-Thought (CoT)** Next, we convert each sequence of vectors $z_0, z_1, \ldots, z_5$ produced by the tasks into a string-based Chain-of-Thought (CoT) representation. Each intermediate state vector $z_i$ is expressed as a comma-separated list of its elements, and the transitions between the states are delimited by "||". This format explicitly captures the step-by-step reasoning process of the model.

For example, given an initial vector $z_0 = [83, 48, 14, 98, 25]$, applying the tasks sequentially yields intermediate states $z_1, z_2, \ldots, z_5$. These states are concatenated into a single string, separated by "||" to represent the full reasoning chain:

```
83,48,14,98,25 || 48,83,98,14,25 || 48,131,229,25,14 || 229,131,48,25,39 ||
            1872,3275,5725,2304,229 || 1403,5725,2304,2075,1872
```

## C.1.3. TRAINING THE SUPERVISOR MODEL

To estimate the information-gain in (3), we train a different LLM, which we refer to as the *supervisor model* $g_{\text{sup}}$. As explained in Section 3.3, this model takes as input the model's CoT reasoning up to any given intermediate step $t$, $X_t^M$, and is fine-tuned to directly predict the ground truth final output $Y$. To this end, we use a special token to separate the model's CoT reasoning and the final output when fine-tuning $g_{\text{sup}}$. At inference time, this special token when appended to the model input serves as an indication for the model to directly predict the final output. In this way $g_{\text{sup}}(X_t^M)$ approximates the conditional distribution $p(Y \mid X_t^M)$.

More specifically, in the toy setup discussed above, consider the following sample for model's CoT:

```
83,48,14,98,25 || 48,83,98,14,25 || 48,131,229,25,14 || 229,131,48,25,39 ||
            1872,3275,5725,2304,229 || 1403,5725,2304,2075,1872
```

For this example, the ground truth final output $y$ is $y =$ "$1403, 5725, 2304, 2075, 1872$" (i.e., the model reached the correct final output in the example above).

For the sample given above, we have that

$$x_0^M = x_0 = \text{``}83,48,14,98,25\text{''}$$
$$x_1^M = \text{``}83,48,14,98,25 \ || \ 48,83,98,14,25 \text{''}$$
$$\vdots$$
$$x_5^M = \text{``}83,48,14,98,25 \ || \ 48,83,98,14,25 \ || \ 48,131,229,25,14 \ ||$$
$$229,131,48,25,39 \ || \ 1872,3275,5725,2304,229 \ ||$$
$$1403,5725,2304,2075,1872\text{''}$$

Next, to construct the data for fine-tuning the supervisor model, we used the special token "$\#|>$" to separate the model's CoT steps $x_i^M$ from the ground truth output $y$. This results in the following 6 training datapoints for the supervisor model:

1. "$83,48,14,98,25 \ \#|> \ 1403,5725,2304,2075,1872$"

2. "83,48,14,98,25 || 48,83,98,14,25 #|> 1403,5725,2304,2075,1872"

    ⋮

5. "83,48,14,98,25 || 48,83,98,14,25 || 48,131,229,25,14 || 229,131,48,25,39
   || 1872,3275,5725,2304,229 || 1403,5725,2304,2075,1872 #|>
   1403,5725,2304,2075,1872"

The above procedure allows us to obtain fine-tuning data for supervisor models separately for each of the 5 different LLMs, $\{\text{LLM}_1, \text{LLM}_2, \ldots, \text{LLM}_5\}$. Next, we train a separate GPT-2 model for each of the 5 different base LLMs.

### C.1.4. ESTIMATING THE INFORMATION-GAIN

Having trained the supervisor model on the data generated above, we evaluate the information-gain on a held-out dataset split. Given a datapoint $(x_i^M, y)$ in the evaluation split, we can estimate the sample-wise information-gain at step $i$ as follows:

- Suppose that the model generation at step $i - 1$, $x_{i-1}^M$ is tokenised as $(t_1, \ldots, t_{n_{i-1}})$ and similarly that $x_i^M$ is tokenised as $(t_1, \ldots, t_{n_i})$. Likewise, suppose that the true output $y$ is tokenised as $(t_1^*, \ldots, t_k^*)$ and we use $< s >$ to denote the separator token (i.e. #|> above).

- Then, to estimate the sample-wise for this datapoint, we estimate the difference:

$$\frac{1}{k} \sum_{j=1}^{k} \log p(t_j^* \mid (t_1, \ldots, t_{n_i}, < s >, t_1^*, \ldots, t_{j-1}^*))$$

$$- \frac{1}{k} \sum_{j=1}^{k} \log p(t_j^* \mid (t_1, \ldots, t_{n_{i-1}}, < s >, t_1^*, \ldots, t_{j-1}^*)).$$

  Here, the supervisor model is trained to estimate the above conditional and therefore we use it to estimate the difference above.

Finally, to estimate the aggregate information-gain (instead of the sample-wise information-gain), we simply compute the average sample-wise gain over the evaluation data split.

### C.1.5. ADDITIONAL RESULTS

In Figures 5 - 7, we present the sample-wise trajectories for 15 randomly chosen prompts leading to incorrect final answers, for the different baselines and LLMs under consideration. Here, any significant drop in the plotted value at a given step could be seen as an indication of an incorrectly executed sub-task. Recall that in our setup, in $\text{LLM}_i$, the CoT step $i$ is executed incorrectly with some probability whereas all other steps are always executed correctly.

Firstly, Figure 5 presents sample-wise information-gain for our method for the five different LLMs. Here, we see that the sample-wise information remains high up until the incorrect step, at which point the information-gain sharply decreases. This suggests that sample-wise information-gain is sensitive to the specific point where the Chain of Thought goes wrong, making it effective at locating reasoning errors.

For the ORM and Math-Shepherd baselines in Figures 6 and 7, we observe that for all LLMs except $\text{LLM}_3$, the plotted metrics drop at the incorrect step. However, for $\text{LLM}_3$, we observe that ORM's probability of correctness drops at step 2 even though the error occurs at step 3. This occurs because, in our setup, the correctness of step 3 is determined directly from the output of step 2. Specifically, recall that in $\text{LLM}_3$, step 3 is executed incorrectly if and only if the output of step 2, $z_2$, has its second component greater than 150, i.e. $z_2[2] > 150$. Therefore, ORM becomes confident after the second step if a CoT is going to lead towards the correct final answer or not.

Similarly, for Math-Shepherd in Figure 7, we observe that the proportion of correct completions remains 0 for $\text{LLM}_3$. This is because for all trajectories plotted, the output of step 2, $z_2$, has its second component greater than 150 and therefore the final answer is incorrect regardless of which step we begin the completions from.

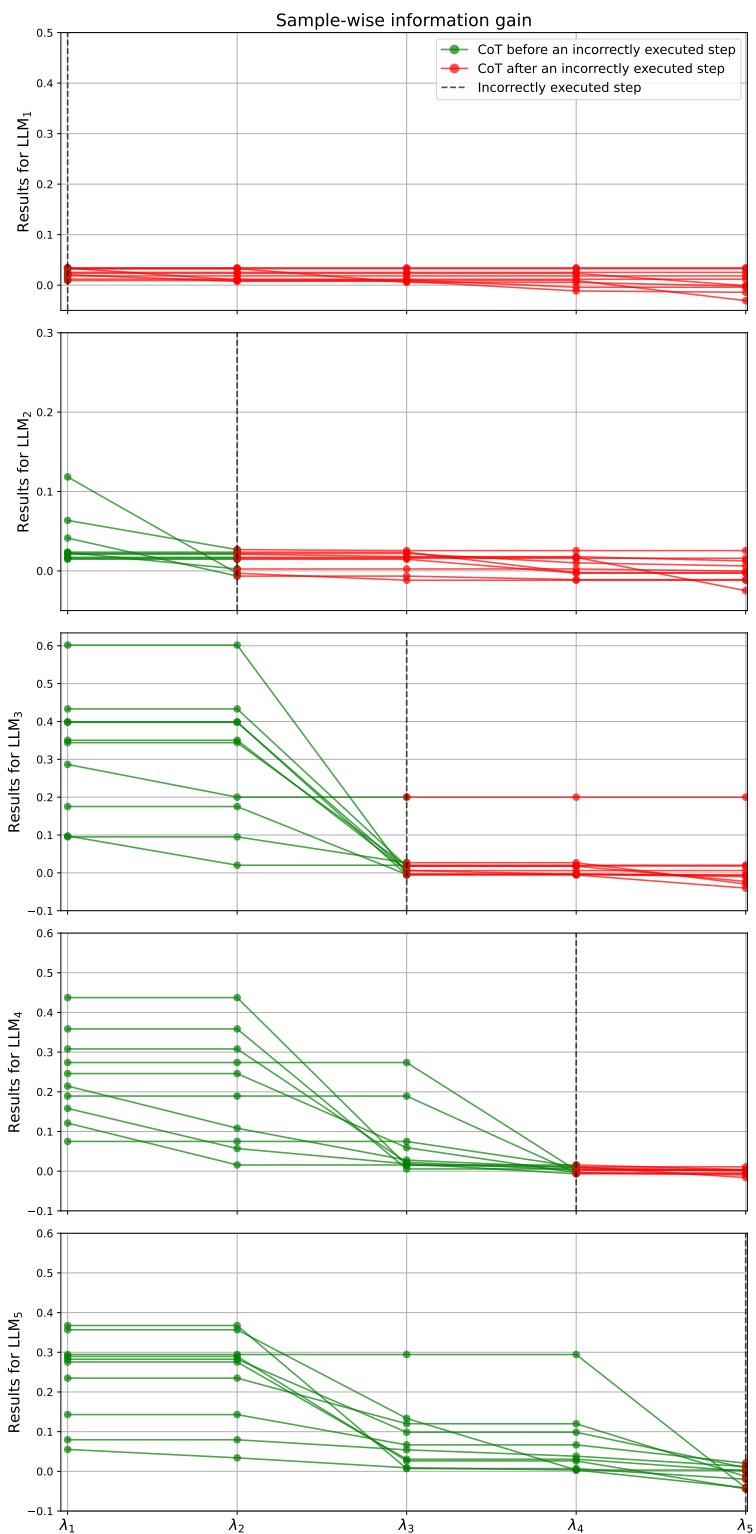

*Figure 5.* Toy data results: Sample-wise information-gain trajectories for 15 randomly chosen prompts with wrong final answers.

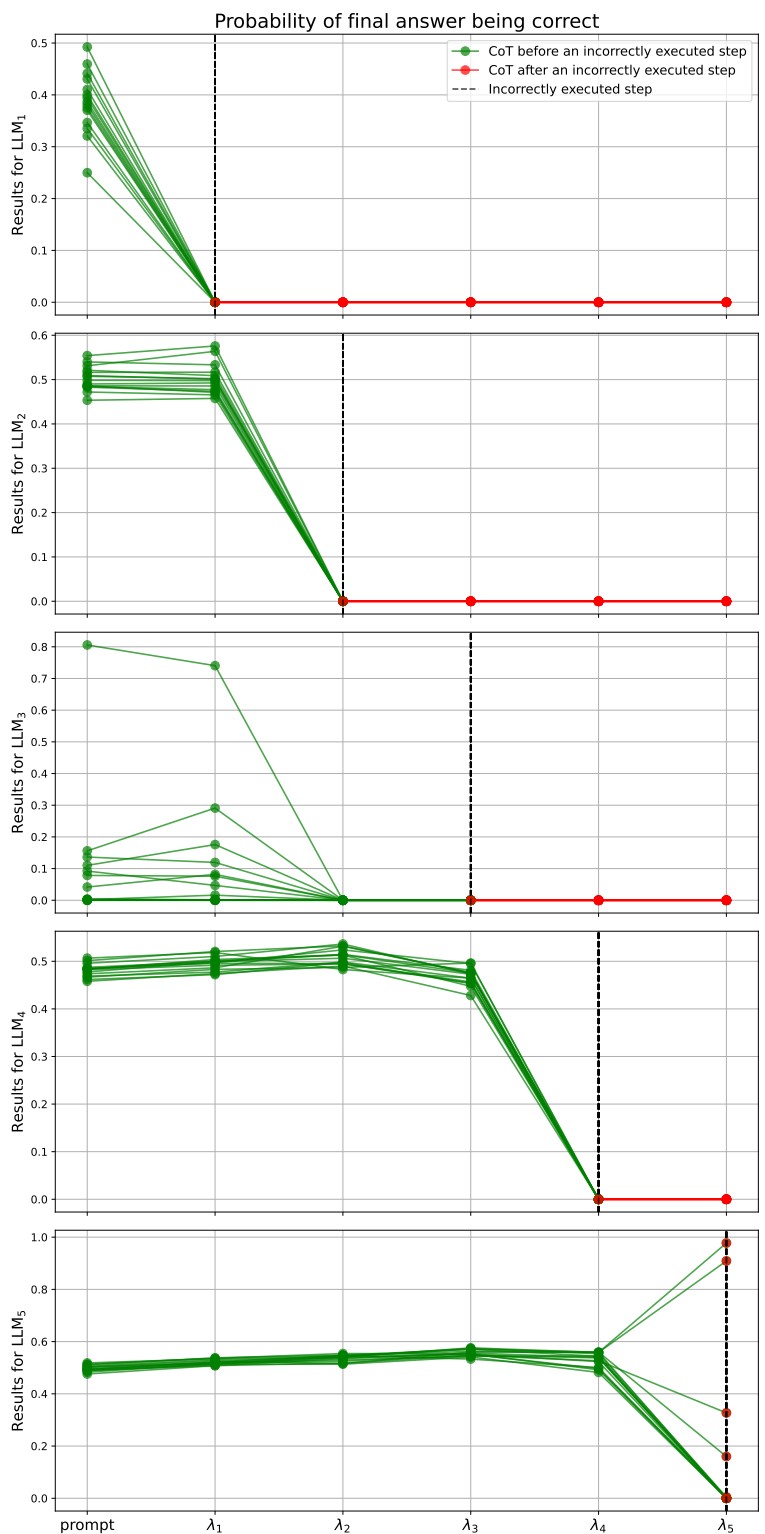

*Figure 6.* Toy data results: ORM's probability of correctness after each step for 15 randomly chosen prompts with wrong final answers

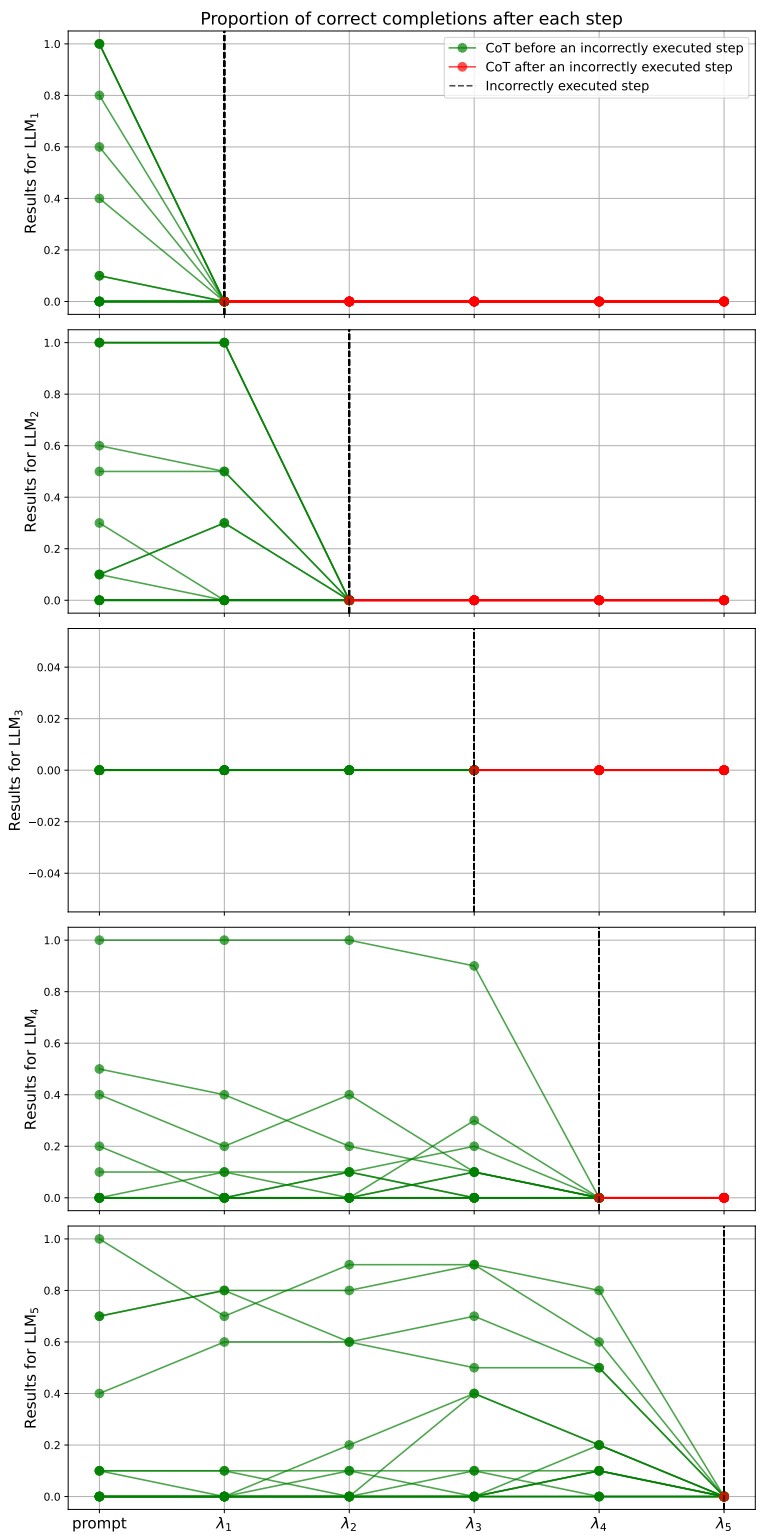

*Figure 7.* Toy data results: Math-Shepherd's proportion of correct completions from each step for 15 randomly chosen prompts with wrong final answers

## C.2. Arithmetic Operations on Llama-3-8b

For this experiment, the prompts used to collect the data follow a specific structure. Each prompt contains two real examples followed by a query with newly sampled values for $x$ and $y$. The format of the prompt is as follows:

```
x = 23, y = 51. Please calculate the following:
1. 3x
2. 2y
3. 3x + 2y
Answer:
1. 3x = 69
2. 2y = 102
3. 3x + 2y = 171

x = 35, y = 60. Please calculate the following:
1. 3x
2. 2y
3. 3x + 2y
Answer:
1. 3x = 105
2. 2y = 120
3. 3x + 2y = 225

x = {x}, y = {y}. Please calculate the following:
1. 3x
2. 2y
3. 3x + 2y
Answer:
```

In the third section, the values of $x$ and $y$ are randomly sampled from a uniform distribution over the range $[1, 10^5)$.

### C.2.1. TRAINING DATA FOR THE SUPERVISOR MODEL

The *supervisor model* plays a crucial role in evaluating the intermediate steps in the Chain-of-Thought (CoT) reasoning. The model is designed to approximate the probability of arriving at the correct final result after any given step in the CoT process. To train this model, we fine-tune it using a dataset composed of generated CoT steps concatenated with the correct final result.

**Model Generation Example:** Consider the following example of a model-generated response:

```
x = 51290.0, y = 90718.0. Please calculate the following:
1. 3x
2. 2y
3. 3x + 2y
Answer:
1. 3x = 153770.0
2. 2y = 181436.0
3. 3x + 2y = 335206.0
```

**Fine-Tuning Data Construction:** The generated outputs are used to construct training examples, where each intermediate step is concatenated with the final correct answer using the separator token '#|>'. For instance, from the example above, the following four training data points are created:

1. "x = 51290.0, y = 90718.0.  Please calculate the following:  1.  3x 2.  2y 3. 3x + 2y Answer:  #|> 3x + 2y = 335306.0"

2. `"x = 51290.0, y = 90718.0.  Please calculate the following:  1.  3x 2.  2y 3. 3x + 2y Answer:  || 1.  3x = 153770.0 #|> 3x + 2y = 335306.0"`

3. `"x = 51290.0, y = 90718.0.  Please calculate the following:  1.  3x 2.  2y 3.  3x + 2y Answer:  || 1.  3x = 153770.0 || 2.  2y = 181436.0 #|> 3x + 2y = 335306.0"`

4. `"x = 51290.0, y = 90718.0.  Please calculate the following:  1.  3x 2.  2y 3. 3x + 2y Answer:  || 1.  3x = 153770.0 || 2.  2y = 181436.0 || 3.  3x + 2y = 335206.0 #|> 3x + 2y = 335306.0"`

Each step concatenates the current state of reasoning with the correct final answer. This process enables the supervisor model to learn the relationship between intermediate steps and the correct final outcome.

Using the dataset generated above, we fine-tune a Llama-3-8b model using Low Rank Adaptation (LoRA) (Hu et al., 2021) as the supervisor model. Finally, the information-gain is computed using the trained model as described in Section C.1.4.

### C.2.2. MATH SHEPHERD RESULTS

The Math-Shepherd approach (Wang et al., 2024b) evaluates how well the model generates intermediate results and completes the reasoning process step-by-step. For a given model generation, we iteratively cut off the chain of reasoning after each step and obtain multiple completions using a completer model (in this case, also the Llama-3-8B model).

Consider the following model generation:

```
x = 51290.0, y = 90718.0. Please calculate the following:
1. 3x
2. 2y
3. 3x + 2y
Answer: 1. 3x = 153770.0, 2. 2y = 181436.0, 3. 3x + 2y = 335206.0
```

In this example, the model completes the full sequence of steps for $x = 51290.0$ and $y = 90718.0$. To assess the robustness of the Chain-of-Thought (CoT) process, we perform the following procedure for the Math Shepherd results:

1. Step-wise Completion: We cut off the generation after each step in the reasoning process. For instance, after computing $3x = 153770.0$, we stop the generation there and generate 10 completions using the Llama-3-8b model.

2. Multiple Completions: At each cut-off point, the Llama-3-8b model is tasked with completing the remaining steps of the chain of reasoning. For each step, 10 independent completions are generated.

3. Proportion of Correct Completions: For each cut-off point, we compute the proportion of correct completions. This proportion gives insight into how likely the model is to complete the remaining steps of reasoning correctly, starting from the intermediate point. For example, after cutting off the reasoning at $3x = 153770.0$, we evaluate how many of the 10 completions successfully compute $3x + 2y = 335306.0$.

In this way, Math-Shepherd quantifies the model's ability to continue reasoning correctly at each intermediate stage.

### C.2.3. ADDITIONAL RESULTS

Figures 8 - 10 present the sample-wise trajectories for 15 randomly chosen prompts leading to incorrect final answers for the different baselines. Here, once again, any significant drop in the plotted value at a given step could be seen as an indication of an incorrectly executed sub-task. Recall that in this setup majority of the errors occur at the final step which involves the addition of $3x + 2y$.

Figure 8 shows the sample-wise information-gain for our method after each step. We see that for most of the plotted trajectories, the sample-wise information-gain remains high until the final step, at which point it drops to values close to or below 0. This shows that our method correctly identifies that the failure predominantly occurs at step 3.

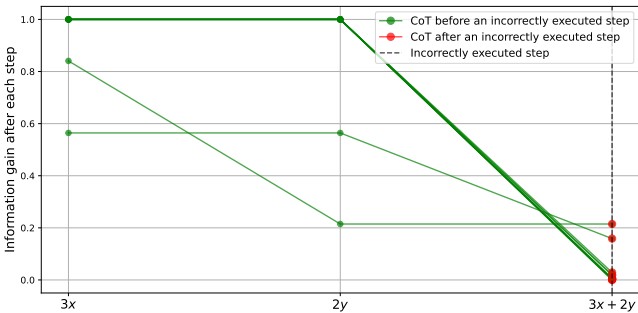

*Figure 8.* Arithmetic operations on Llama-3-8b: Sample-wise information-gain trajectories for 15 randomly chosen prompts with wrong final answers

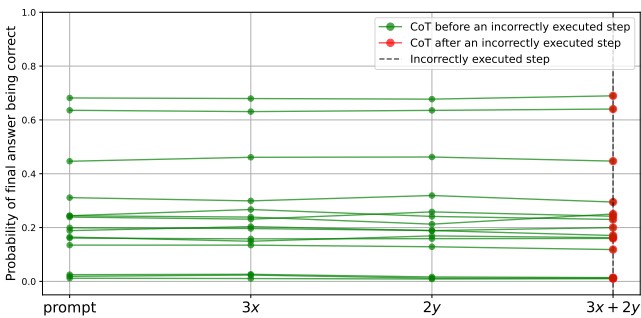

*Figure 9.* Arithmetic operations on Llama-3-8b: ORM's probability of correctness after each step for 15 randomly chosen prompts with wrong final answers

In contrast, Figure 9 shows that the mean probability of correctness for the ORM remains unchanged at each step. This could be explained by Figure 4 in the main text, which suggests that the ORM classifier can predict the correctness of the final output using only the values of $x$ and $y$ available in the prompt. Crucially, the classifier's confidence remains unchanged even as the model's intermediate reasoning steps are added to the input. This means that ORM is unable to distinguish between the model's performance on intermediate reasoning steps.

For Math-Shepherd results shown in Figure 10, most of the trajectories plotted remain constant at 0. In other words, when using Llama-3-8B as the completer model, we observe that for most of the prompts, no completion leads to the correct answer, regardless of which step we begin the completions from. This is likely because, for most of the examples considered in this plot, the $(x, y)$ combination in the prompt has exactly one small value and the other is large (as shown in Figure 4). This also highlights why Math-Shepherd has a high false positive rate.

### C.3. Controlled GSM8K Experiments

In order to understand if our proposed method also works on more textual data, we set out to perform an experiment on the popular GSM8K dataset which has more elaborate prompts compared to the previous experiments. To this end, we follow these steps:

- We first construct the dataset by using the GPT-4 API on the question. This will give us the basis for correct CoTs.

- Next we also again use GPT-4 to label each of the intermediate steps as either using "Addition", "Subtraction", "Division" or "Multiplication".

- With this data in hand, we can now construct our unidentifiable operations. In particular, we again use GPT-4 to rewrite all the CoTs which contain a multiplication into CoTs, where the multiplication is performed wrong and subsequently also the final answer is wrong.

- Finally, we filter the final dataset, where we make sure that for every failed CoT, we have both "multiplication" and

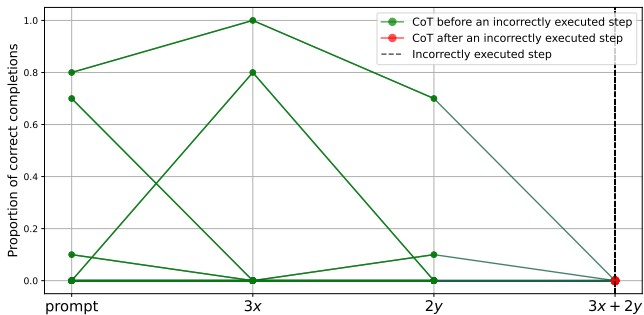

*Figure 10.* Arithmetic operations on Llama-3-8b: Math-Shepherd's proportion of correct completions from each step for 15 randomly chosen prompts with wrong final answers

"subtraction" in the CoT, and for every correct CoT, we do not have these operations. This leads us to a setting where standard methods such as ORM would likely fail as there is no way to distinguish between multiplication and subtraction for the classifier.

### C.3.1. EXAMPLES

Here below are some examples from the dataset, where we start with the prompt, followed by the CoT generations (with a wrong multiplication).

**Prompt 1:**

```
Julie is reading a 120-page book.  Yesterday, she was able to read 12 pages and
today, she read twice as many pages as yesterday.  If she wants to read half of
the remaining pages tomorrow, how many pages should she read?
```

**Model response 1:**

```
Calculate today's pages read by doubling yesterday's pages:  12 * 2 = 46.
Calculate the total pages read so far by adding yesterday's and today's pages:
12 + 46 = 58.  Determine the remaining pages by subtracting the pages read so far
from the total:  120 - 58 = 62.  Calculate the number of pages Julie should read
tomorrow by taking half of the remaining pages:  62 / 2 = 31.
```

**Prompt 2:**

```
In a conference room, 40 chairs with a capacity of 2 people each were arranged in
rows in preparation for the board meeting of a company, whose number of members
was the same as the chairs' capacity.  If 2/5 of the chairs were not occupied,
and the rest each had two people, calculate the number of board members who did
attend the meeting.
```

**Model response 2:**

```
Calculate the total capacity of the conference room:  40 chairs * 2 people per
chair = 134 people.  Calculate the number of chairs not occupied:  40 chairs
* 2/5 = 16 chairs.  Calculate the number of chairs occupied:  Total number of
chairs - Chairs not occupied = 40 chairs - 16 chairs = 24 chairs.  Calculate the
number of people that attended (each occupied chair has 2 people):  24 chairs * 2
people per chair = 63 people.
```

**Prompt 3:**

```
Dan owns an ice cream shop and every sixth customer gets a free ice cream cone.
Cones cost $2 each.  If he sold $100 worth of cones, how many free ones did he
give away?
```

**Model response 3:**

```
Calculate the number of cones sold that were not free by dividing the total
sales by the cost per cone:  100 / 2 = 50.  Calculate the total number of cones
including free ones by assuming every sixth cone is free, so for every 5 sold,
there is 1 free.  Total cones = (5 sold + 1 free) * (cones sold / 5) = 6 * (50 /
5) = 6 * 10 = 72.  Calculate the number of free cones given away:  Total cones -
Cones sold = 72 - 50 = 22.
```

### C.3.2. TRAINING DATA FOR THE SUPERVISOR MODEL

For our supervisor model, we simply use a GPT-2 model that we SFT until convergence and use early stopping based on a held out validation dataset. The training data for this model is composed of generated CoT steps concatenated with the correct final output (as in other experiments).

For example, consider prompt 3 and its response above. For this prompt, the correct final response is 10. Using this prompt, we generate 4 training datapoints for the supervisor model by truncating the response at each step and concatenating the correct final answer using the separator token '#|>'.

1. ```
   Dan owns an ice cream shop and every sixth customer gets a free ice cream
   cone.  Cones cost $2 each.  If he sold $100 worth of cones, how many free
   ones did he give away?  #|> 10
   ```

2. ```
   Dan owns an ice cream shop and every sixth customer gets a free ice cream
   cone.  Cones cost $2 each.  If he sold $100 worth of cones, how many free
   ones did he give away?  || Calculate the number of cones sold that were not
   free by dividing the total sales by the cost per cone:  100 / 2 = 50 #|> 10
   ```

3. ```
   Dan owns an ice cream shop and every sixth customer gets a free ice cream
   cone.  Cones cost $2 each.  If he sold $100 worth of cones, how many free
   ones did he give away?  || Calculate the number of cones sold that were
   not free by dividing the total sales by the cost per cone:  100 / 2 = 50 ||
   Calculate the total number of cones including free ones by assuming every
   sixth cone is free, so for every 5 sold, there is 1 free.  Total cones = (5
   sold + 1 free) * (cones sold / 5) = 6 * (50 / 5) = 6 * 10 = 72 #|> 10
   ```

4. ```
   Dan owns an ice cream shop and every sixth customer gets a free ice cream
   cone.  Cones cost $2 each.  If he sold $100 worth of cones, how many free
   ones did he give away?  || Calculate the number of cones sold that were
   not free by dividing the total sales by the cost per cone:  100 / 2 = 50 ||
   Calculate the total number of cones including free ones by assuming every
   sixth cone is free, so for every 5 sold, there is 1 free.  Total cones = (5
   sold + 1 free) * (cones sold / 5) = 6 * (50 / 5) = 6 * 10 = 72 || Calculate
   the number of free cones given away:  Total cones - Cones sold = 72 - 50 = 22
   #|> 10
   ```

### C.3.3. ESTIMATING THE INFORMATION-GAIN

Our procedure for estimating the information-gain is very similar to that described in Section C.1.4. However, in this setup, there is no fixed ordering of tasks for all prompts. For instance, in some prompts, the first step might be addition while in others it might be multiplication. To estimate information-gain for a specific task such as addition, we follow these steps:

- We first consider all prompts which contain addition as a sub-task.

- Next, for these prompts we estimate the $\mathbb{E}[\log p(Y \mid X_{T_+}^M)]$ term, where $T_+$ denotes the step at which addition is executed.

- Similarly, we estimate the $\mathbb{E}[\log p(Y \mid X_{T_+-1}^M)]$ term, where $T_+ - 1$ denotes the step immediately preceding addition.

- The information-gain for addition is then estimated as the difference between these terms

$$\mathbb{E}[\log p(Y \mid X_{T_+}^M)] - \mathbb{E}[\log p(Y \mid X_{T_+-1}^M)].$$

## C.4. PRM800K Experiments

To further validate our approach, we conduct experiments on the PRM800K dataset (Lightman et al., 2023), which provides step-wise correctness labels for problems derived from the MATH dataset. Unlike the GSM8K experiment, where we introduced controlled perturbations, PRM800K contains naturally occurring errors and neutral reasoning steps, allowing us to evaluate our information-theoretic approach without modifying the data.

### C.4.1. DATASET AND EXPERIMENTAL SETUP

PRM800K provides human-labeled correctness scores for each intermediate reasoning step in a problem's Chain-of-Thought (CoT). Each step is annotated as:

- **Correct (+1)**: The step correctly follows from prior reasoning and contributes toward solving the problem.

- **Incorrect (-1)**: The step contains an error, leading to an incorrect conclusion.

- **Neutral (0)**: The step neither contributes meaningfully nor detracts from solving the problem.

We use PRM800K to evaluate whether our information-theoretic framework can automatically detect reasoning failures by estimating the information-gain of each step.

### C.4.2. TRAINING DATA FOR THE SUPERVISOR MODEL

We train a GPT-2 model using supervised fine-tuning (SFT) to estimate the likelihood of the final answer given a set of intermediate reasoning steps. The training data consists of problem statements and corresponding CoT steps, with the correct final response appended using the separator token '`#|>`'.

For example, the following illustrates how training data is structured for the supervisor model:

1. ```
   How many of the first one hundred positive integers are divisible by 3, 4,
   and 5?  #|> 1
   ```

2. ```
   How many of the first one hundred positive integers are divisible by 3, 4,
   and 5?  || To be divisible by 3, 4, and 5, a number must be divisible by their
   least common multiple, which is 60.  #|> 1
   ```

3. ```
   How many of the first one hundred positive integers are divisible by 3, 4,
   and 5?  || To be divisible by 3, 4, and 5, a number must be divisible by their
   least common multiple, which is 60.  || So, I need to find how many multiples
   of 60 are in the range from 1 to 100.  #|> 1
   ```

4. ```
   How many of the first one hundred positive integers are divisible by 3, 4,
   and 5?  || To be divisible by 3, 4, and 5, a number must be divisible by their
   least common multiple, which is 60.  || So, I need to find how many multiples
   of 60 are in the range from 1 to 100.  || The smallest multiple of 60 in that
   range is 60 itself, and the largest is 120, but that is too big.  #|> 1
   ```

5. ```
   How many of the first one hundred positive integers are divisible by 3, 4,
   and 5?  || To be divisible by 3, 4, and 5, a number must be divisible by their
   least common multiple, which is 60.  || So, I need to find how many multiples
   of 60 are in the range from 1 to 100.  || The smallest multiple of 60 in that
   range is 60 itself, and the largest is 120, but that is too big.  || So, the
   multiples of 60 in that range are 60 and 120/2 = 60 + 30 = 90.  #|> 1
   ```

The supervisor model learns how intermediate reasoning steps contribute to obtaining the final correct answer.

| Method | Learns per step? | Needs labeled CoTs? | Scalable? |
|---|---|---|---|
| ORM | ✗ No | ✗ No | ✓ Yes |
| PRM | ✓ Yes | ✓ Yes | ✗ No |
| IG (Ours) | ✓ Yes | ✗ No | ✓ Yes |

*Table 3.* Comparison of ORM, PRM, and our method based on step-wise learning, labeled CoT dependency, and scalability.

### C.4.3. ESTIMATING INFORMATION-GAIN

Following the procedure in Section C.1.4, we estimate the information-gain of each reasoning step. For a specific step type $\lambda_t$, information-gain is computed as:

$$\mathbb{E}[\log p(Y \mid X_t^M)] - \mathbb{E}[\log p(Y \mid X_{t-1}^M)]$$

where:

- $X_t^M$ is the model's output at step $t$,

- $Y$ is the correct final answer,

- $t$ denotes the step where the operation $\lambda_t$ is applied.

### C.4.4. COMPARISON WITH ORM AND PRM

Table 3 provides a qualitative comparison of our method with:

- **Outcome Reward Modeling (ORM)** (Cobbe et al., 2021; Lightman et al., 2023), which predicts correctness based only on the final answer.

- **Process-based Reward Modeling (PRM)** (Lightman et al., 2023; Uesato et al., 2022), which learns correctness at each intermediate step using labeled CoTs.

Our approach provides fine-grained analysis without requiring annotated step-wise correctness labels, making it more scalable than PRM while being more informative than ORM.

### C.4.5. KEY FINDINGS

Our experiments show:

- **Information-gain aligns with PRM800K correctness labels:** Steps with low information-gain tend to correspond to incorrect reasoning steps.

- **Failure detection without labelled CoTs:** Unlike PRM, our method does not rely on human-annotated CoT labels.

- **Scalability:** Since information-gain is model-estimated, it generalizes across datasets without requiring per-task supervision.

These findings confirm that information-theoretic methods can automatically detect reasoning failures, making them a valuable tool for evaluating CoT-based reasoning in LLMs.

