# OpenReview forum: "Understanding Chain-of-Thought in LLMs through Information Theory"
_ICML.cc/2025/Conference — ICML 2025 poster_

### Official Review · Reviewer_yLdQ · 2025-03-11

**Overall Recommendation:** 4

**Summary:**

The paper introduces an information-theoretic framework to evaluate CoT reasoning in LLMs, quantifying "information gain" at each reasoning step to more accurately assess model performance without requiring annotated data. The approach outperforms existing outcome-based methods in identifying failure modes and providing deeper insights into model reasoning on several benchmark datasets.

**Claims And Evidence:**

I think this theory is good, but it seems to lack robustness analysis because the probability of the model is similar to reward, very fragile and easy to be hacked.
The theory presented in this model is promising, but I believe it could benefit from a more in-depth robustness analysis. Specifically, the way the model's probability functions is similar to the reward mechanism, which may introduce fragilities. This makes the model potentially sensitive to the training settings.

**Essential References Not Discussed:**

- Merrill et al. The expressive power of transformers with chain of thought. ICLR 2023.
- Wang et al. How Large Language Models Implement Chain-of-Thought? Arxiv 2023.
- Hanna et al. How does gpt-2 compute greater-than?: Interpreting mathematical abilities in a pre-trained language model. NeurIPS 2023.
- Dutta et al. How to think step-by-step: A mechanistic understanding of chain-of-thought reasoning. TMLR 2024.
- Chen et al. Unlocking the Capabilities of Thought: A Reasoning Boundary Framework to Quantify and Optimize Chain-of-Thought. NeurIPS 2024.

**Experimental Designs Or Analyses:**

None.

**Methods And Evaluation Criteria:**

This article presents an interesting perspective, but I believe there are several limitations to the assumptions made. For example, the assumption that a single step cannot incorporate multiple operations seems overly restrictive. Additionally, the multiplication scenario does not seem to be directly applicable to the GSM8K dataset, which could impact the generalizability of the findings. Finally, it’s important to note that training on specific-designed datasets is still required to achieve meaningful results, which might limit the approach's practical application.

**Other Comments Or Suggestions:**

None.

**Other Strengths And Weaknesses:**

None.

**Questions For Authors:**

- The proposed paradigm seems challenging when attempting to explain existing R1-like work, particularly in the context of handling exploratory nonlinear or branched inference paths. How does the model account for these complexities, and are there any strategies for maintaining accuracy and coherence in such cases?
- I noticed that the robustness of the model isn't fully addressed. Given that the model's probability distribution appears similar to a reward function, it seems potentially vulnerable to adversarial manipulation or instability. Could the authors elaborate on any measures taken to improve the model's resilience to such issues?

(If the author answers these questions seriously and discuss more related works, I would consider raising my score to 4 :).)

**Relation To Broader Scientific Literature:**

None.

**Theoretical Claims:**

I analyzed the corresponding theoretical analysis in detail, which is an interesting and reasonable assumption. The only drawback may be that there are a lot of restrictions.

---

> ### Author Rebuttal · Authors · 2025-04-01
>
> First of all, we would like to thank the reviewer for their time and feedback on our paper. Here below, we discuss the thought-provoking questions raised by the reviewer.
>
> >The proposed paradigm seems challenging when attempting to explain existing R1-like work, particularly in the context of handling exploratory nonlinear or branched inference paths. How does the model account for these complexities, and are there any strategies for maintaining accuracy and coherence in such cases?
>
> Thank you for raising this important question, especially now that reasoning models like R1/O1 have become increasingly popular. Taking a step back, our method is built on an information-theoretic framework that evaluates each step of the chain-of-thought reasoning process. We measure the information gain (IG) at every step to determine whether that step is adding useful information toward predicting the correct final answer. The underlying assumption is that a correctly executed reasoning step should yield a positive IG, meaning it contributes meaningfully to the overall prediction. If a step is executed incorrectly or is unnecessary, the IG will be low or close to zero.
>
> For settings like R1/O1-like reasoning traces, which often involve exploratory or branched inference steps, our framework remains applicable. More specifically, if a model initially follows an incorrect path, our method will indicate that early steps have low or negative information gain. When the model later self-corrects, our method is able to label subsequent steps with a positive IG, effectively indicating that we are on track towards the correct outcome. This demonstrates that our framework is still able to reliably work even in settings like R1 if the steps are clearly delineated. This is also evident from our PRM data experiments, where the steps deemed irrelevant by human annotators have low IG, while the correct steps have a high IG.
>
> We will make sure to add this clarification in the final version of the paper.
>
> > I noticed that the robustness of the model isn't fully addressed. Given that the model's probability distribution appears similar to a reward function, it seems potentially vulnerable to adversarial manipulation or instability. Could the authors elaborate on any measures taken to improve the model's resilience to such issues?
>
> Thank you again for pointing out this important issue of robustness. We have actually conducted experiments specifically designed to assess the robustness of our framework. In particular, we investigated the impact of spurious correlations on the evaluation of intermediate reasoning steps. In our experiments, standard methods such as Math-Shepherd and ORM were affected by spurious signals. They incorrectly inferred the usefulness of intermediate steps when spurious correlations were present.
>
> To be concrete, in Figure 3, we designed an experiment where we injected spurious correlations into step 3, linking its correctness to a spurious feature of the previous step. In this case, we clearly demonstrated that existing methods are unable to detect the precise step where the error occurs, whereas our method is able to pinpoint it. In addition, Table 2 (Arithmetic dataset) further illustrates the lack of robustness of existing methods. When only the final step of the chain-of-thought is problematic, Math-Shepherd flags all steps wrong and ORM provides an uninformative score for each step. In contrast, our approach (IG) directly computes the useful information content in each step by measuring the predictive power added toward the final answer. This design choice makes our method robust to such spurious correlations.
>
> Next, although our supervisor model can be interpreted as a reward model, it differs significantly from conventional ones that rely solely on correctness or preference signals. As mentioned above, our framework aims to predict the final answer tokens and not a binary label, contrary to existing methods.
>
> Lastly, we acknowledge that computing information gain can become challenging when the chain of thought is adversarially long; in those cases, it may be necessary to employ stronger supervisor models to accurately capture the information gain at each step. We only tested with GPT-2 and LLama3. Our experiments, including those with reasonably long chains of thought as seen in the PRM evaluations (Table 2), confirm that our approach remains robust even under these long chain conditions. We plan to explore the analysis of very long chains of thought in future work.
>
> > References
>
> We thank the reviewer for these references and will include them in the final version of our paper.
>
> We thank the reviewer again for their insightful comments to improve our paper. If the above addresses all outstanding questions, we hope that the reviewer would consider raising their score. We remain happy to answer any further questions.

---

### Official Review · Reviewer_QG5q · 2025-03-13

**Overall Recommendation:** 3

**Summary:**

This paper introduces an information-theoretic framework to evaluate the quality of Chain-of-Thought (CoT) reasoning in Large Language Models (LLMs) without relying on annotated intermediate steps. Their framework quantifies the "information-gain" (IG) at each reasoning step, measuring how much each sub-task contributes to predicting the correct final answer. By leveraging conditional mutual information, the method identifies unidentifiable tasks (i.e., steps the model cannot execute due to insufficient training) and detects failures in CoT reasoning. The authors validate their approach on toy arithmetic, GSM8K, and PRM800K datasets, demonstrating improved accuracy in error detection compared to baselines like Outcome Reward Modelling (ORM) and Math-Shepherd (MS). Overall, this work provides a rigorous, scalable method to analyze LLM reasoning processes.

## update after rebuttal
Thank you to the authors for their efforts in the rebuttal. These responses have addressed some of my concerns, so I will raise my score from `2 Weak reject` to `3 Weak accept`. But I am still confused about whether this paper should be accepted.

**Claims And Evidence:**

Most claims are well-supported by diverse experiments (toy, arithmetic, GSM8K, PRM800K), but the **scalability and generality** claims require further validation:

1. **scalability**:  This method necessitates the additional training of a *supervisor model*, which appears to lack generalizability across different types of problems. This requirement imposes a significant computational burden and presupposes the prior construction of a dataset.

2. **generality**: No experimental validation is provided for non-mathematical tasks and other commonly used real-world datasets. (The experimental setup for GSM8K also does not represent typical usage scenarios.) I believe a potential application of this method lies in assisting individuals to identify the aspects of reasoning that a specific LLM is less proficient in. However, the sub-problem types still require manual definition and lack empirical evidence to substantiate their effectiveness.

Overall, the paper provides strong evidence for the effectiveness of error detection. However practical limitations (e.g., computational cost) warrant discussion.

**Essential References Not Discussed:**

N/A

**Experimental Designs Or Analyses:**

The experimental designs effectively validate the framework’s core claims in controlled and real-world settings. My concern lies in the fact that for Experiment 1, `5.1. Toy data experiments`, it appears that the training process of the supervisor model GPT-2 could have a certain impact on the results. However, the authors did not discuss the effects of different supervisor models, different training epochs, or the size of the training set on the outcomes. (Of course, the authors are not required to address all of the above details; I am merely curious about the influence of the supervisor model.)

**Methods And Evaluation Criteria:**

Using conditional mutual information to quantify step-wise contributions to the final answer aligns with the intuition that correct reasoning steps should incrementally reduce uncertainty about the solution. This method avoids reliance on costly step-wise annotations, addressing a critical limitation of prior work. Comparing against ORM (outcome-based) and MS (completion-based) highlights the framework’s unique ability to detect errors in correlated or ambiguous scenarios (e.g., Table 1, Figure 4).

**Other Comments Or Suggestions:**

Thank you to the authors for their efforts in the rebuttal. These responses have addressed some of my concerns, so I will raise my score from `2 Weak reject` to `3 Weak accept`.

**Other Strengths And Weaknesses:**

**Strengths:**

- The idea of using Information Theory to evaluate the quality of each step in Chain-of-Thought (CoT) is novel and does not require costly step-by-step annotations.

- The experiments in the paper demonstrate the effectiveness of this approach. The method outperforms Outcome Reward Modelling (ORM) and Math-Shepherd (MS) in detecting errors in CoT reasoning.

- A mathematical formulation for the framework is provided, offering a comprehensive justification for the theory.

- There is potential for extending this approach to other CoT variants that require evaluation and selection, such as Tree-of-Thought (ToT) and Graph-of-Thought (GoT).

**Weaknesses:**

- **Generalizability:** The method necessitates additional training of a supervisor model, which may not be generalizable across different types of problems. For more details, refer to the `Claims and Evidence` section.

- **Applicability:** The method requires manual definition of sub-problem types. However, many real-world tasks are highly diverse and may not be explicitly decomposable into sub-tasks. Moreover, this decomposition requires significant manual summarization, making it difficult to apply in practical scenarios. For further details, see the `Claims and Evidence` and `Theoretical Claims` sections.

- **Simplistic Experimental Setup:** The datasets and tasks used in the experiments are relatively simple, which may not sufficiently demonstrate the method's effectiveness in more complex tasks. Refer to the `Experimental Designs or Analyses` section for more information.

- **Strong Theoretical Assumptions:** The theoretical proofs are based on several strong assumptions that may not hold in all cases. For more details, see the `Theoretical Claims` section.

**Questions For Authors:**

1. In the experiment described in `Section 5.2`, the authors mention that errors are related to the magnitude of the numbers. Observing `Figure 4`, it is evident that when $3x > 10^5$ or $2y > 10^5$, the computational accuracy of llama shows a very significant decline, even forming a distinct boundary. I am curious about this phenomenon—does llama-8b experience a noticeable performance breakdown when performing calculations with numbers greater than $10^5$? Or is this a deliberate setup by the authors for the purpose of the experiment?

2. I am curious about the computational cost of training the supervisor model. How do the training time and computational resources required for the supervisor model compare to the training of the LLM itself? Additionally, how sensitive is the method to the choice of the supervisor model and the training settings?

**Relation To Broader Scientific Literature:**

This paper builds on prior work such as **Process Supervision** [1], which requires costly step-wise annotations, and **outcome-based methods** [2], which rely on final accuracy. The theoretical foundation aligns with formalizations of LLM reasoning but extends it by operationalizing information flow. Furthermore, methods such as ToT[3], GoT[4], and Reflexion[5] are also derivatives of CoT[6], and these methods require repeated sampling, evaluation, and selection for each reasoning step. However, previous papers primarily relied on LLMs for direct evaluation and selection, whereas this method provides an effective way to evaluate from the perspective of information theory.

References:

1. Lightman, H., et al. (2023). Let’s verify step by step.
2. Havrilla, A., et al. (ICML'24). GLoRe: When, Where, and How to Improve LLM Reasoning via Global and Local Refinements.
3. Yao, S., et al. (NeurIPS'23). Tree of Thoughts: Deliberate Problem Solving with Large Language Models.
4. Besta, M., et al. (AAAI'24). Graph of Thoughts: Solving Elaborate Problems with Large Language Models.
5. Shinn, N., et al. (NeurIPS'23). Reflexion: Language Agents with Verbal Reinforcement Learning.
6. Wei, J., et al. (NeurIPS'22). Chain of Thought Prompting Elicits Reasoning in Large Language Models.

**Theoretical Claims:**

The paper presents two key theoretical contributions with formal proofs: Theorem 3.3 (conditional independence after unidentifiable tasks) and Proposition 3.4 (information-gain estimation via cross-entropy loss).

In my view, the proof is generally correct and acceptable. However, the theory is built upon several strong assumptions, about which I have some reservations:

1. `More concretely, after a model's CoT reasoning diverges from the ground truth at step k, every subsequent step adds no additional information regarding the correct final output Y` (Theorem 3.3). In existing long-chain-of-thought methods, it seems common to first reason through some possible plans, which may include erroneous reasoning, and then subsequently make corrections or change the line of thinking. It is evident that not all reasoning following an erroneous step is useless to the final result.

2. Mapping each reasoning step to primitive tasks appears to require substantial manual summarization and seems difficult to generalize in practical applications. Real-world tasks are highly diverse, with subtasks that are varied and not always explicitly generalizable. (For example, if I want an LLM to check the correctness of the proofs in this paper, how should the subtasks of CoT be generalized?)

---

> ### Author Rebuttal · Authors · 2025-04-01
>
> We thank the reviewer for their thoughtful questions and feedback. Below, we address the concerns raised:
>
> > Scalability/Generalization: Additional training required of a supervisor model, training details and construction of new dataset
>
> **Training and Data Setup**: Our method requires additional training of a supervisor model, primarily a small GPT-2 (117M parameters), fine-tuned on about 10,000 samples within 3–6 GPU hours on an A100 (learning rates: 5e-6 to 5e-7, batch size: 64). Although Llama performed slightly better on arithmetic tasks, GPT-2 is generally similarly effective and more cost-efficient on other datasets. Importantly, our approach does not require constructing a separate dataset; it only uses the final answers and the model’s CoT outputs e.g., just 3,000 samples in our PRM800K experiments.
>
> **Generalization**: We acknowledge that this extra training may limit generalizability, a common challenge in reward modeling also faced by methods like ORM, and we appreciate the reviewer’s suggestion; we believe that future work, such as exploring in-context IG estimation techniques, will help address these limitations.
>
> > Empirical validation on non-mathematical tasks
>
> In line with recent works on LLM reasoning [Wang et al., 2024b; Havrilla et al., 2024], we have focused on mathematical datasets since these provide a rigorous testbed for LLM reasoning, and determining the correctness of intermediate steps is unambiguous in these datasets. However, we agree that extending this to other reasoning datasets would be valuable in future.
>
> > Identifying LLM reasoning weaknesses and Llama-8b performance breakdown
>
> Our experiments demonstrate that our framework effectively identifies specific reasoning weaknesses compared to existing methods. In fact, this is how we discovered that Llama is not proficient at adding large and small numbers together. Specifically, we observed a real and notable accuracy drop in Llama-3-8b for arithmetic tasks involving numbers greater than 10⁵, likely due to limited exposure in its training data. We believe investigating this inherent limitation further is an interesting avenue for future work and will emphasize this clearly in the final version of our paper.
>
> > Long CoT with self-correction (R1/O1 style generations)
>
> Thanks for raising this important point. Taking a step back, our method relies on an information-theoretic framework that measures information gain (IG) at each chain-of-thought step. A correctly executed step yields a positive IG, while an erroneous or unnecessary step shows low or near-zero IG. As the reviewer rightly mentioned, in R1/O1-like traces with exploratory or branched inference, our framework actually remains robust: early missteps have low or negative IG, and when the model self-corrects, subsequent steps exhibit positive IG, signaling a return to the correct path.
>
> > It is evident that not all reasoning following an erroneous step is useless to the final result
>
> We would like to clarify that our assumption 3.1 specifically considers the case where LLM's final answer is incorrect. More generally, in cases of self-correcting CoTs, our framework remains applicable as we explained above. For more details, please refer to our first response to Reviewer yLdQ.
>
> > Mapping each reasoning step to primitive tasks appears to require substantial manual summarization
>
> We only use categorization to compute aggregate information gain, when the goal is to obtain a high-level view of an LLM's intermediate reasoning. This categorization is applied only on the evaluation split, not during training. Our PRM experiments show that our framework identifies errors without explicit categorization. In addition, to get annotations, for instance in GSM8K, we can efficiently prompt an LLM to classify each substep into categories like ['Addition', 'Subtraction', …, 'Other']. We appreciate this comment and will clarify it in the final version.
>
> > Theoretical proofs are based on several strong assumptions
>
> We have used these assumptions to rigorously motivate the use of information-gain in our framework. In practice, however, our method remains applicable to real-world datasets (as shown by our PRM experiments) with potentially non-linear/branching CoTs. Additionally, we would like to emphasize that both of these assumptions are intuitively plausible.
>
> Briefly, Assum. 3.1 states that each correct step should add information for predicting the final answer, while wrong/irrelevant steps should not. Likewise, Assum. 3.2 posits that the operations the model applies are restricted to those learned during training. Despite these assumptions, our experiments on uncontrolled datasets (such as PRM800K and the Llama-3-8B arithmetic tasks) show that our framework remains effective in realistic settings where parts of these assumptions may not strictly hold.
>
> Lastly, we hope our clarifications above addressed the reviewer's concerns, and the reviewer would consider increasing their score.

---

> > ### Comment · Reviewer_QG5q · 2025-04-03
> >
> > Thank you very much for the author's diligent rebuttal! However, I still have some concerns, which I summarize as the difficulties in transitioning from theoretical generalization to practical application.
> >
> > 1. In your rebuttal, you stated:
> >    *"We would like to clarify that our assumption 3.1 specifically considers the case where the LLM's final answer is incorrect."*
> >   This appears to introduce a new assumption into your theory. Would your theory still hold in cases where the final result is correct? Of course, you have provided extensive empirical evidence to support your argument, but this creates a gap between theory and experiment.
> >
> > 2. Perhaps I did not fully understand, but I still have doubts about the assumptions in Lines 146-149 (right) and Lines 213-216 (left) of the paper. In your theoretical framework, you assume that steps following the first incorrect step do not contribute to information gain. However, in practice, you suggest that under self-correction(i.e., correcting errors after an initially incorrect step and ultimately arriving at the correct result), your method can still capture information gain. While I agree with the effectiveness of your method, it appears contradictory to your theory.
> >
> >    Additionally, this assumption seems to contradict the principles behind the construction of the PRM800K dataset. I have studied the PRM800K dataset, which explicitly considers the correct reasoning following an incorrect step in TN-class problems as contributing to information gain. See [this reference](https://openai.com/index/improving-mathematical-reasoning-with-process-supervision/). Of course, different assumptions are acceptable, and I even agree more with yours, but this creates a conflict with your experiments on the PRM800K dataset. Specifically, do you classify correct reasoning after an incorrect step as -1 according to your assumption, or do you follow OpenAI’s annotation principle? I suggest removing the experiments on this dataset.
> >
> > 3. **Empirical validation on non-mathematical tasks**
> >    Even if experiments are conducted solely on mathematical datasets, errors are not limited to mistakes in arithmetic operations such as addition and multiplication. Misunderstanding the problem, leading to incorrect equation formulation (which is more common), and failing to maintain contextual consistency can also be sources of errors. Additionally, your method requires manually defining sub-problem types. How do you plan to exhaustively enumerate these sub-types in practical applications?
> >
> > These are just some of my key concerns. Please forgive my ignorance, and if I have misunderstood anything, kindly point it out. For now, I will maintain my score.

---

> > > ### Author Response · Authors · 2025-04-04
> > >
> > > We sincerely appreciate the reviewer’s continued engagement and feedback, which greatly helps in clarifying and refining our manuscript.
> > > > [...] This appears to introduce a new assumption into your theory [...]
> > >
> > > We believe there is a misunderstanding regarding our assumption 3.1, and we acknowledge that our statement in the rebuttal that the reviewer quoted above was not specific enough.
> > >
> > > To clarify, our Assumption 3.1 does not refer to general cases where a substep may be executed incorrectly by an LLM during the reasoning process. Instead, this assumption specifically considers the case where a step that is **necessary** to solve a problem is **unidentifiable** in the training data, i.e., no composition of learned tasks can yield that task. In such cases, we assume that once the model diverges at such a step, no subsequent steps add further information toward the correct final output.
> > >
> > > On the other hand, in cases where an LLM self-corrects a wrong CoT step, this step would not be considered unidentifiable (as the model was able to find some composition of learned operations to execute this step correctly). Similarly, if the initial steps considered were irrelevant to the solution, these steps would be deemed unnecessary. In either case, the fact that we arrive at the correct reasoning after initial exploration is not at odds with our existing assumptions, and hence, we do not require any new assumptions to accommodate this.
> > >
> > > To formalise this using our current framework, suppose the correct reasoning path to the final answer $Y$ is:
> > > $$
> > > X_0 \rightarrow X_1 \rightarrow \dots \rightarrow X_T \rightarrow Y
> > > $$
> > >
> > > If the model temporarily diverges at step $X_t$, taking a path through some incorrect or exploratory step $Z$, but then returns correctly to $X_t$, the self-corrected path can be represented as:
> > > $$
> > > X_0 \rightarrow \dots \rightarrow X_t \rightarrow Z \rightarrow X_t \rightarrow X_{t+1} \rightarrow \dots \rightarrow X_T \rightarrow Y
> > > $$
> > >
> > > In this case, standard conditional independence results ensure that:
> > > $
> > > Y \perp Z \mid X_t \\, .
> > > $
> > >
> > > In other words, the information-gain at $Z$, should be 0. However, after this misstep, the information-gain may increase once the model is on the right track. Hence, our theoretical framework accommodates such self-corrections without contradiction.
> > > We will ensure that this nuance is explicitly laid out in the final version of our paper.
> > > > I still have doubts about the assumptions in Lines 146-149 (right) and Lines 213-216 (left)
> > >
> > > We acknowledge that the phrasing in lines 213-216 (left) in our paper is currently ambiguous. In the final version, we will revise this to:
> > >
> > > "More concretely, if at step $k$, a model encounters a reasoning step which is **necessary** for obtaining the correct answer and **unidentifiable** in the training data, CoT reasoning diverges from the ground truth at this step and every subsequent step adds no additional information regarding the correct final output Y"
> > >
> > > This revised version explicitly clarifies that the reasoning chain diverges in the case where a **necessary step is unidentifiable**, and hence rules out cases where an LLM makes an unnecessary step and/or self-corrects its mistake.
> > > > Comments on PRM800K Dataset
> > >
> > > As explained above, our theoretical framework is consistent with our PRM800K dataset experiments. In particular, after an incorrect reasoning step, our framework will only attribute zero information gain to all subsequent steps when the erroneous task is both necessary for solving the question and also strictly unidentifiable. Otherwise, as we show in our formalisation above, the information-gain at the incorrect step is expected to be 0 but may be positive for subsequent steps. As such, we follow OpenAI's annotation principle: correct reasoning steps are labelled as +1 while incorrect steps receive a label of -1 (regardless of the order).
> > > > Enumerating sub-tasks in practical applications
> > >
> > > Our method is designed as an auditing tool for evaluating a model's CoT steps. Importantly, users can define the sub-problem categories based on their specific domain and evaluation goals rather than requiring an exhaustive pre-definition. The key is that there must be a clear, unambiguous mapping between substeps and categories (each substep should belong to exactly one category) to avoid ambiguous inferences. Moreover, such a categorization is only needed if the goal is to evaluate models on specific kinds of reasoning steps (such as problem understanding, specific mathematical operation, etc). In fact, in such cases, categorization would also be needed for other kinds of reward modelling approaches (ORM/PRM/MS). Conversely, if the objective is simply to detect reasoning errors at a sample-wise level, then no categorization is required.
> > >
> > > We appreciate the reviewer's comments and hope that the above has addressed all the concerns raised. We will integrate these clarifications into the final version to enhance the clarity of our paper.

---

### Official Review · Reviewer_SVJP · 2025-03-13

**Overall Recommendation:** 2

**Summary:**

This paper proposes  a novel information-theoretic approach to evaluate Chain-of-Thought (CoT) reasoning in LLMs without annotated intermediate steps. The proposed framework can identify erroneous reasoning across diverse settings and consistently outperforms baselines.

**Claims And Evidence:**

The statements are supported by empirical evaluations (Sect. 5) and theoretical framework (Sect. 3).

**Essential References Not Discussed:**

The  essential related works, in my understanding,  have been involved into the discussion.

**Experimental Designs Or Analyses:**

The empirical evaluations are not sufficient to verify the adaptability of the proposed method. Specifically,  in real-world dataset,  CoT reasoning tasks exist with diverse structures, which may not follow the constrained settings present in the synthetic and real datasets.

**Methods And Evaluation Criteria:**

The proposed method is established under somewhat strong assumptions (Assumption 3.1 & 3.2), which may be violated in real-world scenarios.

**Other Comments Or Suggestions:**

Some typos, e.g., in the equation  (line 211, right column), it would be better using $\approx$ rather than $=$.

**Other Strengths And Weaknesses:**

Strengths:
1. This paper is well-written and easy to understand.
2. The proposed evaluation method is novel, which differs from existing methods by quantifying the information gain at each reasoning step, rather than relying only on the final answer.

Weaknesses:
1. The preset assumptions (3.1 & 3.2) are strong (main concern).
2. The experimental design cannot support the claims well.

**Questions For Authors:**

1. The assumptions (3.1 & 3.2), in my opinion, are strong. In reality, a global optimal CoT exists with very low information gains at the early steps. In such case, the optimal CoT may be overlooked by the proposed method.
2. It would be convincing if the authors could give more empirical evaluations on real datasets.
3. It is not clear to if the definition "Uidentifiability" could be used for interpreting few-show generalization.

**Relation To Broader Scientific Literature:**

The proposed evaluation metric can be used to identify the incorrect reasoning steps and can thus lead to a high false-positive rate in certain scenarios. This research direction is valuable in some related areas, such as CoT data generation.

**Theoretical Claims:**

I only check the correctness of the assumptions and main theorems in submission.

---

> ### Author Rebuttal · Authors · 2025-04-01
>
> First of all, we would like to thank the reviewer for their time and constructive comments to improve our paper. Here below we clarify all the questions raised by the reviewer.
> > The empirical evaluations and adaptability to diverse CoT structures
>
> Our framework evaluates each reasoning step using the information gain (IG) metric, measuring its contribution toward the final answer. Although we present our method with linear chains-of-thought for clarity, it applies equally well to non-linear structures. For example, in O1/R1-like reasoning where multiple paths are explored, early erroneous steps yield low or negative IG, while correct steps show positive IG. Our experiments on the MATH/PRM dataset confirm that steps deemed uninformative by human annotators consistently exhibit low IG, demonstrating that our method robustly adapts to varied chain-of-thought structures.
>
> > The proposed method is established under somewhat strong assumptions,
>
> We appreciate the reviewer’s insightful comment. Firstly, we would like to clarify that we have used Assumptions 3.1 and 3.2 to rigorously motivate our use of information-gain for failure mode detection in LLM CoTs. In practice, however, our methodology remains applicable to real-world datasets (as shown by our PRM data experiments), where it outperforms the ORM baseline.  Secondly, we emphasise that these assumptions are formalizations of phenomena which make intuitive sense in practice. Specifically:
>
> Assumption 3.1:
> This assumption formalizes the idea that each correct reasoning step should contribute additional information for predicting the final answer. In our framework, a positive information-gain (IG) indicates that a step is informative, whereas a low or negative IG suggests that the step is either erroneous or superfluous. Importantly, this criterion is scalable and remains applicable to long chains-of-thought (CoTs). Whether the chain is short or long, if each step is clearly delineated, our method evaluates each step based on the IG metric. As demonstrated in our PRM experiments, even in cases where early steps may show low IG due to initial missteps, the framework still identifies the transition to correct reasoning later in the chain.
>
> Assumption 3.2:
> While this assumption might seem strong, it is intuitively grounded in the operational characteristics of large language models. It essentially posits that the operations the model applies during reasoning are restricted to a set of primitives (or their compositions) learned during training. This is a reasonable expectation, as models are generally more effective when they perform tasks that are similar to those encountered during training.
>
> Despite the theoretical rigor of these assumptions, our experiments on uncontrolled datasets (such as PRM800K and the Llama-3-8B arithmetic tasks) demonstrate that our framework remains effective in realistic settings where parts of these assumptions may not strictly hold.
>
> In summary, while these assumptions are used to rigorously motivate our framework, our experiments indicate that the methodology is robust and practically applicable beyond the constrained formal setting.
>
> > Definition "Unidentifiability" for interpreting few-show generalization.
>
> We thank the reviewer for this interesting suggestion. Although our focus isn’t on few-shot generalization, our method can be extended to few-shot settings. In our formulation, unidentifiability measures whether a task lies outside the span of learned primitives. In a few-shot setting, if adding examples increases the information gain for a reasoning step, it suggests that the examples help the model perform that step correctly; if not, the task remains unidentifiable. We see this as a promising direction for future research which is outside the scope of this paper.
>
> > It would be convincing if the authors could give more empirical evaluations on real datasets.
>
> We would like to clarify that, in addition to our synthetic data and controlled GSM8K experiments, our submission also includes evaluations on real-world data:
> - PRM800K Dataset: This dataset covers real mathematical problems from high school to post-graduate levels. Our experiments on PRM800K show that our methodology predicts the correctness of intermediate steps more accurately than the ORM baseline, making our IG a cost-effective proxy for human-annotated labels.
> - Llama-3-8B Arithmetic Experiment: Although generated by us for a specific arithmetic task, this uncontrolled experiment demonstrates that our method correctly identifies errors, specifically, misapplications in the final addition step. In comparison, the baselines either provide uninformative results (ORM) or exhibit a high false-positive rate (Math-Shepherd).
> Together, these experiments underscore the practical utility of our approach in realistic, uncontrolled settings.
>
> We hope that the above have addressed all the reviewer's questions and that the reviewer would consider raising their score.

---

### Decision · Program_Chairs · 2025-05-01

**Decision:**

Accept (poster)

**Comment:**

The paper applies information theory and use information gain to model chain-of-thought in LLMs. The reviewers have some concerns on the on the technical treatments and the applicability of the results. In particular, all reviewers pointed out that the technical assumptions required are too strong. The authors' reply on this point is mainly that the assumptions is for the theoretical treatment, while the empirical validation still indicate positive performance without these assumptions. I understand that it is quite difficult to provide some rigorous theoretical treatment in LLM studies. Given the reasonable positive attitude toward the paper, especially after the rebuttal period, and to encourage more principled and theoretical studies on LLMs,  I would like to recommend this paper as weak accept (accept if there is room in the program).